# ADDRESSING SAMPLE INEFFICIENCY IN MULTI-VIEW REPRESENTATION LEARNING

## ABSTRACT

Non-contrastive self-supervised learning (NC-SSL) methods like BarlowTwins and VICReg have shown great promise for label-free representation learning in computer vision. Despite the apparent simplicity of these techniques, researchers must rely on several empirical heuristics to achieve competitive performance, most notably using high-dimensional projector heads and two augmentations of the same image. In this work, we provide theoretical insights on the implicit bias of the BarlowTwins and VICReg loss that can explain these heuristics and guide the development of more principled recommendations. Our first insight is that the orthogonality of the features is more important than projector dimensionality for learning good representations. Based on this, we empirically demonstrate that low-dimensional projector heads are sufficient with appropriate regularization, contrary to the existing heuristic. Our second theoretical insight suggests that using multiple data augmentations better represents the desiderata of the SSL objective. Based on this, we demonstrate that leveraging more augmentations per sample improves representation quality and trainability. In particular, it improves optimization convergence, leading to better features emerging earlier in the training. Remarkably, we demonstrate that we can reduce the pretraining dataset size by up to 4x while maintaining accuracy and improving convergence simply by using more data augmentations. Combining these insights, we present practical pretraining recommendations that improve wall-clock time by 2x and improve performance on CIFAR-10/STL-10 datasets using a ResNet-50 backbone. Thus, this work provides a theoretical insight into NC-SSL and produces practical recommendations for improving its sample and compute efficiency.

## 1 INTRODUCTION

Unsupervised representation learning, i.e., learning features without human-annotated labels, is critical for progress in computer vision. Modern approaches, grouped under the *self-supervised learning (SSL)* umbrella, build on the core insight that *similar* images should map to nearby points in the learned feature space. Current SSL methods can be broadly categorized into contrastive and non-contrastive algorithms. While both categories aim to learn the desired features using "positive" samples, which refer to different augmentations of the same image, they diverge in using "negative" samples. Contrastive methods use augmentations obtained from completely different images as negative samples to avoid the trivial solution of mapping all samples to the same point in the feature space (i.e., representational collapse). But, this necessitates an elaborate sampling scheme and huge batch sizes. Non-contrastive methods, on the other hand, eliminate the need for negative samples altogether and instead rely on regularizing the feature space to avoid representational collapse.

A prominent subgroup among non-contrastive SSL methods is the family of Canonical Correlation Analysis (CCA) algorithms, which includes BarlowTwins (Zbontar et al., 2021) and VICReg (Bardes et al., 2021). These methods aim to enforce orthogonality among the learned features in addition to learning to map similar images to nearby points in feature space and have been shown to achieve competitive performance on benchmark computer vision datasets. These methods have become the preferred strategy for representation learning in several domains due to the lack of need for negative samples and their simple formulation. However, despite the apparent simplicity of their loss functions, the behavior of this family of algorithms is not well understood. Therefore, researchers often use empirically-driven heuristics to design successful applications, such as using (i) a high-

dimensional projector head or (ii) two augmentations per image. Although these heuristics help in practice, their theoretical underpinnings are unclear.

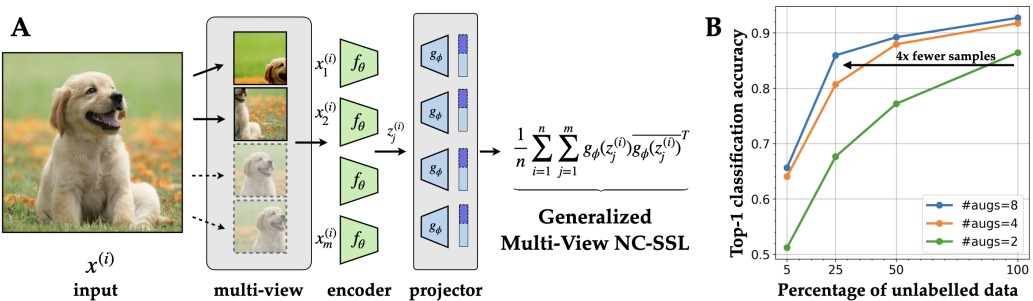

Figure 1: Existing SSl algorithms make design choices often driven by heuristics. (A) We investigate the theoretical underpinnings of two choices (i) the number of augmentations (ii) the dimensionality of the projector. (B) We show that generalized NC-SSL algorithm with multiple augmentations, low-dimensional projectors outperform existing heuristics while using $\sim 4\times$ fewer unlabelled samples.

Alongside relying on heuristics and researchers' intuition for design, existing SSL algorithms are extremely data-hungry. In particular, state-of-the-art algorithms often rely on large-scale datasets (Russakovsky et al., 2015) or data engines (Oquab et al., 2023) to achieve good representations. While this strategy works exceptionally well in natural-image settings, its application is limited in other critical domains, such as medical imaging, where the number of samples is scarce.

With these challenges in mind, the primary focus of this work is making progress toward establishing theoretical foundations underlying the family of non-contrastive SSL algorithms (NC-SSL) with an eye toward sample efficiency. In particular, we analyse the BarlowTwins and VICReg losses and show that they implicitly learn the data similarity kernel that is defined by the chosen augmentations. We find that learning the data similarity kernel is helped by greater orthogonality in the projector outputs and more data augmentations. As such, increasing the orthogonality of the projector output eliminates the requirement for a high-dimensional projector head, and increasing the number of data augmentations decreases the number of unique samples required. Our theoretical analysis establishes a principled grounding for the role of multiple augmentations, and the sufficiency of low-dimensional projectors, together outlining a framework for improving the sample-efficiency of NC-SSL while maintaining representation quality.

We empirically verify our theoretical insights using the popular ResNet-50 backbone on benchmark datasets, CIFAR-10 and STL-10. Strikingly, we show that our multi-augmentation approach can learn good features even with a quarter of the number of samples in the pretraining dataset. As such, this suggests that SSL training can be done with smaller datasets and opens interesting questions in the design of performance enhancing transformations. In summary, our core contributions are:

- **Eigenfunction interpretation:** We demonstrate that the loss functions of the CCA family of non-contrastive SSL algorithms are equivalent to the objective of learning eigenfunctions of the augmentation-defined data kernel.

- **Role of heuristics:** We provide a mechanistic explanation for the role of projector dimensionality and the number of data augmentations, and empirically demonstrate that low-dimensional projector heads are sufficient and using more augmentations leads to learning better representations.

- **Data efficient NC-SSL:** Leveraging the convergence benefits of the multi-augmentation framework, we demonstrate that we can learn good features with significantly smaller datasets (upto 25%) without harming downstream performance.

## 2 PRELIMINARIES

We start by formally defining the unsupervised representation learning problem for computer vision. In particular, we assume access to a dataset $\mathcal{D} = \{x_1, x_2, ..., x_n\}$ with $x_i \in \mathbb{R}^p$ consisting of unla-

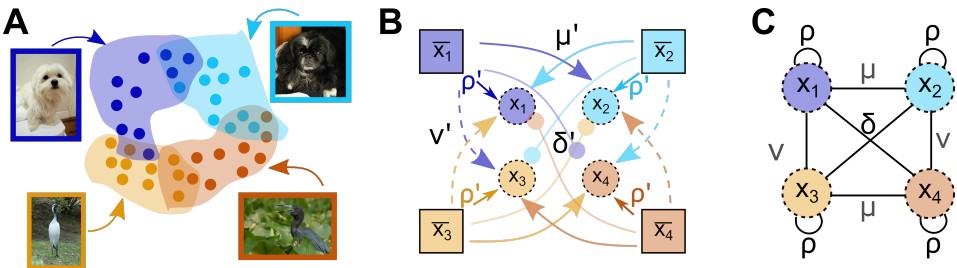

Figure 2: Schematic of augmentation graph. (A) Augmentations from each image span a region in the image space which could overlap with the augmentation span of other images. (B) An augmentation graph schematic that uses probabilities to characterize the interactions among augmentation spans of different instances.

beled instances (often natural images), and the objective is to learn a $d$-dimensional representation ($d < p$) that are useful across multiple downstream applications. We focus on learning the parameters of a deep neural network $f_\theta \in \mathcal{F}_\Theta$, using the multi-view invariance SSL framework, wherein multiple views of an example are used to optimize pretraining loss function, $\mathcal{L}_{pretrain}(f_\theta, \mathcal{D})$

**Non-Contrastive Self-Supervised Learning** (NC-SSL) algorithms impose invariance to *data-augmentations*, which are used to define multiple views of the same image while imposing certain regularization on the geometry of the learned feature space. More generally, both families can be thought of as decomposing $\mathcal{L}_{pretrain}$ into two terms (i) $\mathcal{L}_{invariance}$ : to learn invariance to data augmentations and (ii) $\mathcal{L}_{collapse}$ to prevent collapsing the feature space to some trivial solution with no discriminative power.

$$\mathcal{L}_{pretrain} := \mathcal{L}_{invariance} + \beta \mathcal{L}_{collapse} \qquad (1)$$

where $\beta$ denotes a hyperparameter that controls the importance of the collapse-preventing term relative to the invariance term.

This formulation separates instance-level attributes invariant to augmentations, highlighting the semantic information of the instance. The ideal feature space is less sensitive to varying attributes and more sensitive to semantic ones, facilitating generalization to new examples. Understanding the interplay between pretraining loss and preserved attributes is critical for time and compute-efficiency.

**Data Augmentation graph** was introduced by HaoChen et al. (2021) to analyze contrastive losses, like SimCLR (Chen et al., 2020). Briefly, we define a graph $\mathcal{G}(\mathcal{X}, \mathcal{W})$ with the vertex set $(\mathcal{X}, \rho_X)$ comprising the result of all possible data augmentations from each sample in a dataset (could be infinite when continuous augmentation functions are used) and $\mathcal{W}$ denoting the adjacency matrix. Let $x_0$ be an image in $\mathcal{X}$, and let $z = M(x_0)$ be a random data augmentation of the image, $x_0$. We define the probability density of reaching $z$ from $x_0$ via a choice of mapping $M$:

$$p(z \mid x_0) = \mathbb{P}(z = M(x_0)), \qquad (2)$$

Since mapping are not generally invertible (e.g. crops), observe that $p(x_0 \mid z) \neq p(z \mid x_0)$. Using this definition, we now define the strength of edge between nodes $x$ and $z$ of the augmentation graph as the joint probability of generating augmentations $x, z$ from the same image $x_0 \sim \rho_X$. Formally,

$$w_{xz} := \mathbb{E}_{x_0 \sim \rho_X} \left[ p(x \mid x_0) p(z \mid x_0) \right] \qquad (3)$$

It is worth noting that the magnitude of $w_{xz}$ captures the relative similarity between $x$ and $z$. A higher value of $w_{xz}$ indicates a higher likelihood that both patches came from the same image.

HaoChen et al. (2021) showed that optimizing a functionally equivalent form of the SimCLR loss, termed the spectral contrastive loss ($\mathcal{L}_c$, essentially learns features whose covariance structure matches the adjacency matrix of the augmentation graph.

$$\mathcal{L}_c \propto \| ZZ^T - \bar{\mathcal{W}} \|_F^2 \qquad (4)$$

where $Z$ denotes the output of the neural network, $\bar{\mathcal{W}}$ denotes the degree-normalized adjacency matrix and $\|.\|_F$ denotes the Frobenius norm operator. This perspective implies that the features learned by a contrastive SSL framework would align with the top eigenvectors of $\bar{\mathcal{W}}$. As observed

by HaoChen et al. (2021), all rotations of $Z$ that don't change its span define an equivalence class of solutions to the above optimization problem and make no difference for the downstream generalization of a linear probe. Based on this insight, we define a notion of equivalence among learned feature spaces.

**Definition 2.1.** Let $F(x) = (f_1(x), \ldots f_d(x))$ be a $d$-dimensional feature vector (a vector of functions). Define the subspace

$$V = V(F) = \{h : X \to \mathbb{R} \mid h(x) = w \cdot F(x), \quad w \in \mathbb{R}^d\} \tag{5}$$

to be the span of the components of $F$. Given an $n$-dimensional feature vector, $G(x) = (g_1(x), \ldots, g_n(x))$ we say the features $G$ and $F$ are equivalent, if $V(F) = V(G)$.

## 3    DATA AUGMENTATION KERNEL PERSPECTIVE OF NON-CONTRASTIVE SSL

Following the previous section, we will now present an augmentation kernel perspective of BarlowTwins and VICReg losses. Specifically, we show that the these losses are equivalent to the optimization problem of learning eigenfunctions of the augmentation-defined data covariance kernel. Subsequently, we argue that using a high-dimensional projector yields better overlap with the top eigenvectors of the data augmentation kernel at initialization as compared to a low-dimensional projector. Therefore, our analysis suggests using a stronger orthogonalization constraint during optimization for lower-dimensional projectors to ensure that features learned are equivalent to those learned with high-dimensional projectors. Furthermore, we also argue that using more number of augmentations improves our estimate of the augmentation-defined data covariance kernel, thereby aiding the eigenfunction optimization problem. Therefore, our analysis suggests using an averaging operator with more data augmentations to better estimate the true augmentation kernel.

### 3.1    FEATURES IN TERMS OF DATA AUGMENTATION KERNELS

We will define two notions of the data augmentation kernel. Given two images, $x, z$, the first kernel, which we call the forward data augmentation covariance kernel, is given by

$$k^{DAF}(x, z) = \mathbb{E}_{x_0 \sim \rho_X} \left[ \frac{p(x \mid x_0)}{\rho(x)} \frac{p(z \mid x_0)}{\rho(z)} \right] \tag{6}$$

This covariance kernel measures the similarity between $x, z$ in terms of how likely they are to be reached from $x_0$, weighted by the distribution of $x_0$. Note that this is indeed the edge strength between nodes $x, z$ in the augmentation graph. We can also define a (backwards) data augmentation covariance kernel which reverses the roles of $(x,z)$ and $x_0$:

$$k^{DAB}(x, z) = \mathbb{E}_{x_0 \sim \rho_X} \left[ \frac{p(x_0 \mid x)}{\rho(x_0)} \frac{p(x_0 \mid z)}{\rho(x_0)} \right] \tag{7}$$

The goal of SSL is to learn features that preserve the covariance kernel structure (imposed by this choice of mapping $M$) (Dubois et al., 2022). Therefore, we want to define a loss which determines *vector features*, $F : X \to \mathbb{R}^d$, which factor a data augmentation kernel $k^{DA}(x, z) = F(x)^\top F(z)$. Doing this directly is prohibitively data intensive at scale, since it involves a search over data augmented images. However, since the covariance kernels are PSD, they define a Reproducing Kernel Hilbert space (RKHS). This allows us to apply Mercer's theorem to find vector features as in Deng et al. (2022a;b); Pfau et al. (2018).

The construction of features using Mercer's theorem goes as follows. Given a PSD data augmentation kernel, $k^{DA}$, define the $T_k$ operator, which takes a function $f$ and returns its convolution with the data augmentation kernel.

$$T_k f(x) = \mathbb{E}_{z \sim \rho_X}[k(z, x) f(z)] \tag{8}$$

We will also make use of the the following operator,

$$T_M f(x) = \mathbb{E}_{x_0 \sim M(x)}[f(x_0)] = \sum_{x_0} [p(x_0 \mid x) f(x_0)] \tag{9}$$

which averages the values of function, $f$, over the augmented images $x_0 = M(x)$ of the data, $x$.

Since the operator $T_k$ is compact and positive, it has a spectral decomposition consisting of eigenfunctions $\phi_i$ and corresponding eigenvalues $\lambda_i$. Using these eigenpairs, we can define the (infinite sequence of square summable) spectral features, $G : X \to \ell_2$, (where $\ell_2$ represents square summable sequences), by

$$G(x) = (\sqrt{\lambda_1}\phi_1(x), \ldots, \sqrt{\lambda_d}\phi_d(x), \ldots) \tag{10}$$

Then, Mercer's theorem gives

$$k^{DA}(x, z) = G(x) \cdot G(z) \tag{Mercer}$$

and ensures that the inner product is finite. These are the desired features, which factor the kernel. However, computing the eigenfunctions of $T_k$ is costly. Instead we propose an alternative using the more efficient operator $T_M$. Both operators lead to equivalent features, according to Definition 2.1.

**Theorem 3.1.** *Let $G(x)$ be the infinite Mercer features of the backward data augmentation covariance kernels, $k^{DAB}$. Let $F(x) = (f_1(x), f_2(x), \ldots, f_k(x))$ be the features given by minimizing the following data augmentation invariance loss*

$$L(F) = \sum_{i=1}^{N_k} \|T_M f_i - f_i\|^2_{L^2(\rho_X)}, \quad \text{subject to} \quad (f_i, f_j)_{\rho_X} = \delta_{ij} \tag{11}$$

*which includes the orthogonality constraint. Then, $V(F) \subset V(G)$, $V(F) \to V(G)$ as $N_k \to \infty$.*

The idea of the proof uses the fact that, as linear operators, $T_{k^{DAB}} = T_M^\top T_M$ and that $T_{k^{DAF}} = T_M T_M^\top$. Then we use spectral theory of compact operators, which is analogue of the Singular Value Decomposition in Hilbert Space, to show that eigenfunctions of $T_M^\top T_M$ operator are the same as those obtained from optimizing $L(F)$. A similar result can be obtained using $k^{DAF}$ and $T_M^\top$.

Note that $L(F)$ is the constrained optimization formulation of the BarlowTwins loss. Furthermore, $L(F)$ with the additional constraint that $(f_i, f_i) \geq \gamma \, \forall i \in \{1, 2 \ldots N_k\}$ is the constrained optimization formulation of the VICReg loss.

### 3.2 COROLLARY 1: LOW-DIMENSIONAL PROJECTORS ARE SUFFICIENT

While BarlowTwins and VICReg frameworks have advocated the use of high-dimensional projectors to facilitate good feature learning on Imagenet, our kernel perspective challenges this notion. Since the intrinsic dimensionality of Imagenet is estimated to be $\sim 40$ (Pope et al., 2020), it is not unreasonable to expect that the span of desired features would be of similar dimensionality. It is, thus, intriguing that these frameworks mandate the use of an $\sim 8192 - d$ projector head to capture the intricacies of corresponding data augmentation kernel. This discrepancy can be explained by observing the learning dynamics of a linearized model under the BarlowTwins loss optimization (Simon et al., 2023). These dynamics reveal that initializing the projection weight matrix in alignment with the eigenfunctions of the data kernel retains this alignment throughout the learning process. Notably, a high-dimensional projector is more likely to have a greater span at initialization compared to its low-dimensional counterpart, increasing the likelihood of overlap with the relevant eigenfunctions. We hypothesize that it is possible to rectify this issue by using a stronger orthogonalization constraint for low-dimensional projectors, thereby rendering them sufficient for good feature learning.

### 3.3 COROLLARY 2: MULTIPLE AUGMENTATIONS IMPROVE OPTIMIZATION

Theorem 3.1 implies that the invariance loss optimization would ideally entail using the $T_M$ operator, thereby requiring many augmentations for each sample $x$. Using only two augmentations per sample yields a noisy estimate of $T_M$, yielding spurious eigenpairs (Vershynin, 2010) (see Appendix). These spurious eigenpairs add stochasticity to the learning dynamics, and hinder the alignment of the learned features with the eigenfunctions of the data kernel (Simon et al., 2023). We hypothesize that improving this estimation error by increasing the number of augmentations could ameliorate this issue and improve the speed and quality of feature learning.

Increasing the number of augmentations (say $m$) in BarlowTwins and VICReg comes with added compute costs. A straightforward approach would involve computing the invariance loss for every

pair of augmentations, resulting in $\mathcal{O}(m^2)$ operations. However, Theorem 3.1 proposes an alternative method that uses the sample estimate of $T_M$, thereby requiring only $\mathcal{O}(m)$ operations. Both these strategies are functionally equivalent (see Appendix), but the latter is computationally more efficient. In summary, Theorem 3.1 establishes a mechanistic role for the number of data augmentations, paving the way for a computationally efficient multi-augmentation framework:

$$\widehat{L}(F) = \mathbb{E}_{x \sim \rho_X} \left[ \sum_{i=1}^{N_k} \sum_{j=1}^{m} \|\overline{f_i(x)} - f_i(x_j)\|_{L^2(\rho_X)}^2 \right], \quad \text{subject to} \quad (f_i, f_j)_{\rho_X} = \delta_{ij} \quad (12)$$

where $\overline{f_i(x)} = \frac{1}{m} \sum_{j=1}^{m} f_i(x_j)$ is the sample estimate of $T_M f_i(x)$.

## 4 EXPERIMENTS

In our experiments, we seek to serve two purposes (i) provide empirical support for our theoretical insights and (ii) present practical primitives for designing efficient self-supervised learning routines. In summary, with extensive experiments across learning algorithms (BarlowTwins, VICReg) and training datasets (CIFAR-10/STL-10), we establish that

- **low-dimensional projectors** as sufficient for learning *good representations*.
- **multi-augmentation** improves downstream accuracy and convergence.
- multi-Augmentation **improves sample efficiency** in SSL pretraining, i.e. recovering similar performance with significantly fewer unlabelled samples.

**Experiment Setup**: We evaluate the effectiveness of different pretraining approaches for non-contrastive SSL algorithms using image classification as the downstream task. Across all experiments, we use linear probing with Resnet-50 as the feature encoder backbone. On CIFAR-10, all models are pretrained for 100 epochs, while STL-10 models are pretrained for 50 epochs. All runs are averaged over 3 seeds, and errorbars indicate standard deviation. Other details related to optimizers, learning rate, etc. are presented in the Appendix.

### 4.1 SUFFICIENCY OF LOW-DIMENSIONAL PROJECTORS

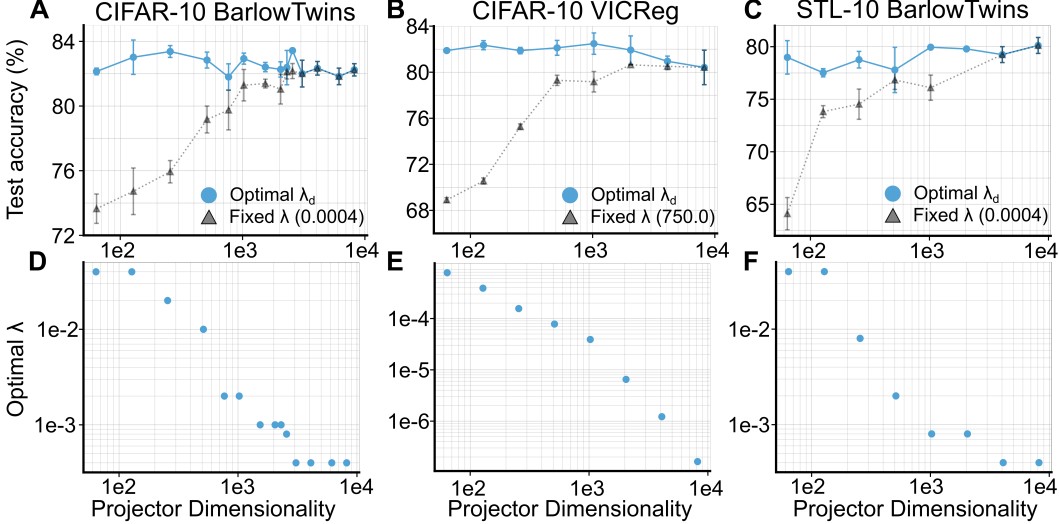

Figure 3: Low-dimensional projectors are sufficient for good feature learning. We demonstrate that using a higher orthogonality constraint ($\lambda$ for D, F and $\lambda_{eff} = \frac{1}{d\lambda}$ for E) for lower projector dimensionality can achieve similar performance over a wide range of projector dimensions ($d$).

Existing works recommend using high-dimensional MLPs as projectors (e.g., d=8192 for Imagenet in Zbontar et al. (2021); Bardes et al. (2021)), and show significant degradation in performance for a fixed redundancy coefficient ($\lambda$). To reproduce this result, we run a grid search to find the optimal coefficient ($\lambda_{8192}^*$) for $d = 8192$ and show that performance progressively degrades for lower $d$ if the same coefficient $\lambda_{8192}^*$ is reused for $d \in \{64, 128, 256, 512, 1024, 2048, 4096, 8192\}$.

Our insights in Section 3.2 suggest low-dimensional projectors should recover similar performance with appropriate orthogonalization. To test this, we find the best $\lambda$ by performing a grid search independently for each $d \in \{64, 128, 256, 512, 1024, 2048, 4096, 8192\}$. As illustrated in Figure 3, low-dimensional projectors are indeed sufficient. Strikingly, we also observe that the optimal $\lambda_d \propto 1/d$, is in alignment with our theoretical insights.

> **Recommendataions:** Start with low-dimensional projector, using $\lambda = \mathcal{O}(\frac{1}{d})$, and sweep over $(pdim = d, \lambda = \mathcal{O}\left(\frac{1}{d}\right))$ if needed.

## 4.2 MULTIPLE AUGMENTATIONS IMPROVE PERFORMANCE AND CONVERGENCE

Although some SSL pretraining approaches, like SWaV, incorporate more than two views, the most widely used heuristic in non-contrastive SSL algorithms involve using two views jointly encoded by a shared backbone. In line with this observation, our baselines for examining the role of multiple augmentations use two views for computing the cross-correlation matrix.

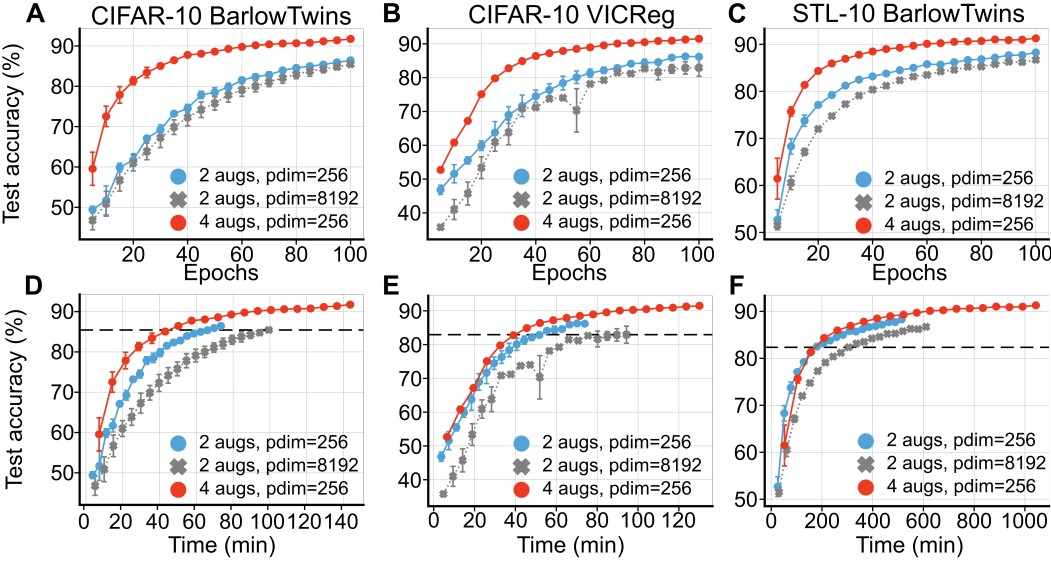

Figure 4: Using multiple augmentations improves representation learning performance and convergence. (A-C) Across BarlowTwins and VICReg for CIFAR-10 and STL-10 pretraining, using 4 augmentations instead of 2 helps improve performance. (D-F) Although the 4-augmentations take longer for each epoch, its performance still trumps the 2-augmentation version of the algorithm at the same wall clock time.

To understand the role of multiple augmentations in pretraining in light of the augmentation-kernel interpretation, we propose Equation (12), which generalizes Barlow-Twins and VICReg to the multi-augmentation setting. In particular, for $\#augs \in \{2, 4, 8\}$, we pretrain Resnet-50 with the generalized NC-SSL loss for 100 epochs on CIFAR-10 and 50-epochs for STL-10. Building on the insight from the previous section, we use a 256-dimensional projector head for all experiments.

In Figure 4, we track the downstream performance of the pre-trained models across training epochs, i.e., we extract features from intermediate checkpoints and train a linear classifier on top of the features. Here, we use the linear evaluation protocol as outlined by Chen et al. (2022). Figure 4(A-C), shows that pretraining with multiple augmentations outperforms the 2-augmentation baseline. Furthermore, we observe that the four-augmentation pre-trained models converge faster (both in

terms of the number of epochs and wall-clock time) than their two-augmentation counterparts (see Figure 4(D-F)).

> **Recommendatation:** Using multiple augmentations ( $> 2$ ) with the generalized NC-SSL loss is likely to improve convergence as well as downstream accuracy.

### 4.3 SAMPLE EFFICIENT MULTI-VIEW LEARNING

Data Augmentation can be viewed as a form of data-inflation, where the number of training samples is increased by a factor of $k$ (for $k$ augmentations). In this section, we examine the role of multi-augmentation in improving sample efficiency. In particular, we are interested in understanding if the same performance can be achieved with a fraction of the pretraining dataset.

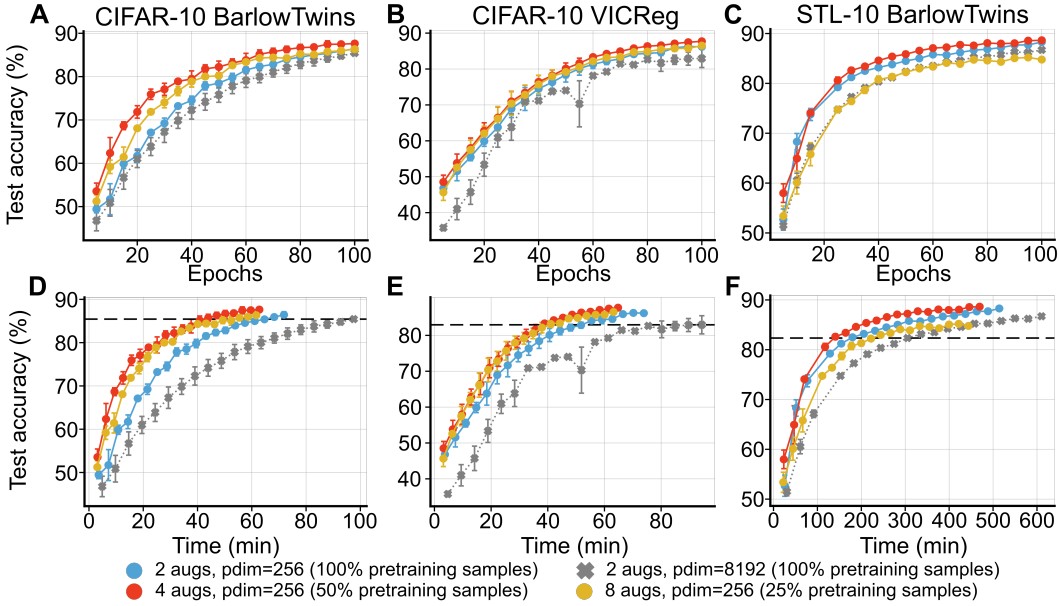

Figure 5: Multi-augmentation improves sample efficiency, recovering similar performance with significantly less number of unique samples in the pretraining dataset. Across BarlowTwins and VICReg pretraining on CIFAR-10 and STL-10, for the same effective dataset size ($\#augs \times \#unique\_samples$), using more patches improves performance at the same epoch (A-C) or wall clock time (D-F). However, there exists a tradeoff wherein doing more data augmentations fails to improve performance in the very low data regime.

To examine the relation between the number of augmentations and sample efficiency, we fixed the effective size of the inflated dataset. This is achieved by varying the fraction of the unique samples in the pretraining dataset depending on the number of augmentations $k \in \{2, 4, 8\}$, e.g. we use 1/2 the dataset for 4 views. We then evaluate the performance of the pre-trained models on the downstream task, where the linear classifier is trained on the same set of labeled samples. Strikingly, Figure 5 shows that using multiple augmentations can achieve similar (sometimes even better) performance with lesser pretraining samples, thereby indicating that more data augmentations can be used to compensate for smaller pretraining datasets.

> **Recommendation:** Use more, diverse augmentations for sample efficient pretraining.

## 5 RELATED WORK

**Self-Supervised Pretraining** requires significant compute resources, and with the lack of a unified theoretical framework, most practitioners rely on empirical heuristics. The SSL cookbook (Balestriero et al., 2023) provides a comprehensive summary of several such widely adopted practices. While recent advances in SSL theory explore learning dynamics in linear (or shallow) models

(Tian et al., 2020; 2021), with a focus on understanding dimensionality collapse (Jing et al., 2021), the theoretical underpinnings of most "recipes" essential for good feature learning, are missing.

**Contrastive SSL** has received more theoretical attention, owing to its connection with metric learning and noise contrastive estimation (Li et al., 2021; Balestriero & LeCun, 2022; Johnson et al., 2023). In particular, HaoChen et al. (2021) provide a theoretical framework for the SimCLR loss from an augmentation graph perspective, which leads to practical recommendations. Subsequently, Garrido et al. (2022) establish a duality between contrastive and non-contrastive learning objectives, further bridging the gap between theory and practice.

**Non-Contrastive SSL** algorithms have comparitively scarce theoretical foundations. In prior work (Agrawal et al., 2022; Garrido et al., 2022) find that with modified learning objectives, low-dimensional projectors are sufficient for good downstream performance. Similarly, previous works have demonstrated notable performance boosts when using a multi-patch framework in contrastive (Dwibedi et al., 2021) and non-contrastive SSL. However, the theoretical basis for the benefits and trade-offs of either low-dimensional projectors or multiple augmentations is unclear.

**Deep Learning theory** has made significant strides in understanding the optimization landscape and dynamics of supervised learning (Advani et al., 2020). In recent work, Simon et al. (2023) use a simpler formulation of the BarlowTwins loss and investigate the learning dynamics in linearized models for the case when the invariance and orthogonalization losses have equal penalties.

## 6 DISCUSSION

**Summary**: Our work presents a fresh theoretical analysis that sheds light on the implicit bias of non-contrastive SSL algorithms. We use these insights to unravel the impact of key design heuristics and offer practical recommendations that improve convergence while maintaining accuracy (on CIFAR-10/STL-10). We also show that the multi-augmentation framework can be used to learn good features from fewer unique samples in the pretraining dataset, simply by improving the estimation of the data augmentation kernel.

**Pareto Optimal SSL** In the context of sample efficiency, training a model using two augmentations with different fractions of the dataset leads to a natural Pareto frontier, i.e. training on the full dataset achieves the best error but takes the most time (**Baseline (2-Aug)**). Our extensive experiments demonstrate that using more than two augmentations improves the overall Pareto frontier, i.e. achieves better convergence while maintaining accuracy (**Multi-Aug**). Strikingly, as shown in Figure 6, we observe that for a target error level, we can either use a larger pretraining dataset or more augmentations. Therefore, the number of augmentations can be used as a knob to control the sample efficiency of the pretraining routine.

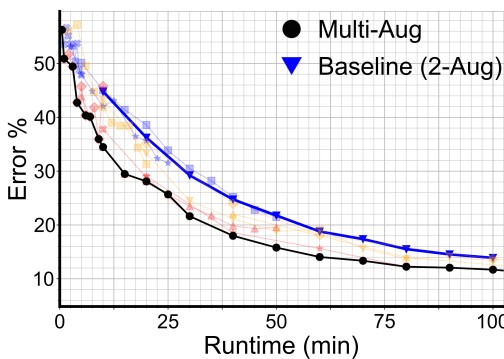

Figure 6: Using $> 2$ augmentations with a fraction of the dataset improves overall Pareto frontier, speeding runtime by upto $\sim 2\times$.

**Open Questions**: Looking ahead, it would be exciting to extend this analysis to other categories of SSL algorithms, such as Masked AutoEncoders (MAE). Furthermore, our insights provide opportunities to explore sample-efficient methods that rely on less data, which is particularly important in critical domains such as medical imaging, where data is often scarce and expensive to obtain.

**Limitations** Our algorithm relies on multiple views of the same image to improve estimation of the data-augmentation kernel. Although this approach does add some extra computational overhead, it significantly speeds up the learning process. We can explore the possibility of making the current design more computationally efficient to further improve it.

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

# A  ICLR Reviewer Response: Additional Experiments and Discussion

## A.1  Longer Pretraining

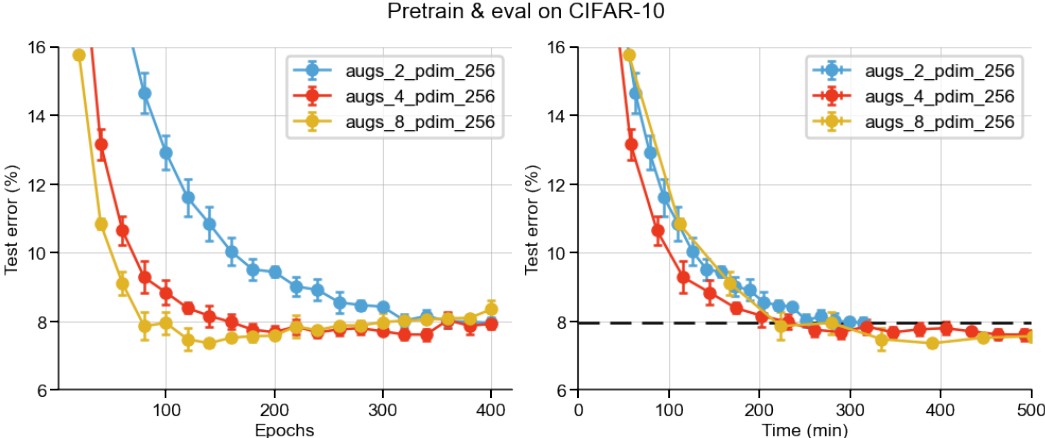

Figure 7: BarlowTwins pretraining on full CIFAR-10 dataset for 400 epochs.

| Algorithm | Best accuracy | Best accuracy @ epoch |
|---|---|---|
| Barlow-Twins (2-augs) w/ pdim=256 | 92.04 +/- 0.16 | 400 |
| Barlow-Twins (4-augs) w/ pdim=256 | 92.39 +/- 0.17 | 340 |
| Barlow-Twins (8-augs) w/ pdim=256 | 92.64 +/- 0.10 | 140 |

Table 1: BarlowTwins pretraining on full CIFAR-10 dataset at 400 epochs (with early stopping)

## A.2  SwAV-like augmentations for compute efficient multi-augmentation framework

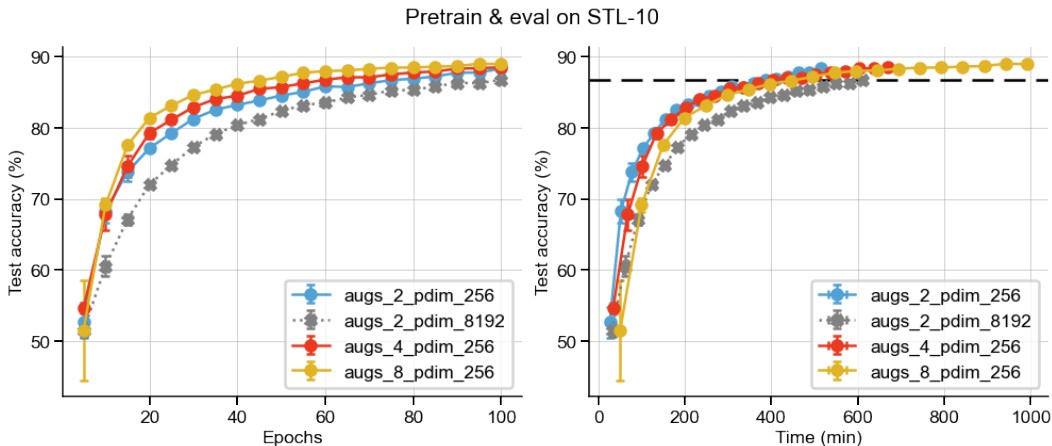

Figure 8: BarlowTwins pretraining on full STL-10 dataset for 100 epochs using SwAV-like augmentations. Specifically, the 2-augmentations setting uses two views that are $64 \times 64$, whereas the 4 (or 8) augmentation setting uses additional two (or six) augmentations that are $32 \times 32$.

## A.3 ON TRAINING WITH FULL DATASET WITH 4/8 AUGMENTATIONS

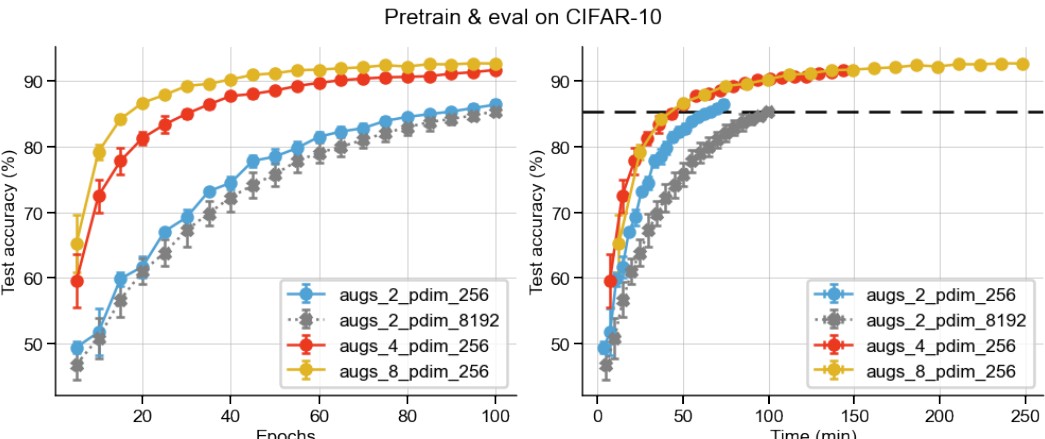

Figure 9: BarlowTwins pretraining on full CIFAR-10 dataset.

| Algorithm | #augs=2 | #augs=4 | #augs=8 |
|---|---|---|---|
| Barlow-Twins w/ pdim=256 | 86.43 +/- 0.72 | 91.73 +/- 0.16 | 92.71 +/- 0.19 |
| Barlow-Twins w/ pdim=8192 | 85.44 +/- 0.54 | 91.40 +/- 0.32 | 92.40 +/- 0.13 |

Table 2: BarlowTwins pretraining on full CIFAR-10 dataset at 100 epochs

# B  HILBERT SPACE OF FUNCTIONS

## B.1  FUNCTIONS AND INNER PRODUCT SPACE

**Definition B.1.** Given $X, \rho_X$, and $f, g : X \to \mathbb{R}$, define the $L^2(\rho_X)$ inner product and norm, respectively,

$$(f, g)_{\rho_X} = \int f(x)g(x)d\rho_X(x), \quad \|f\|_{\rho_X}^2 = (f, f)_{\rho_X} \tag{13}$$

Define

$$L^2(\rho, X) = \left\{ f : X \to \mathbb{R} \mid \|f\|_{\rho_X}^2 < \infty \right\}$$

to be the (equivalence class) of functions with finite $\rho_X$ norm.

## B.2  SPECTRAL THEORY

In this section we quote the relevant (abstract) Hilbert Space theory.

**Definition B.2** (Spectral Operator). Given orthogonal functions, $\Phi = (\phi_i)_{i \in I}$ in $L^2(\rho_X)$, and non-negative $\Lambda = (\lambda_i)_{i \in I}$, with $\|\Lambda\|_2^2 = \sum_{i \in I} \lambda_i^2 < \infty$. Call $(\Phi, \Lambda)$ a spectral pair and define the corresponding spectral operator by

$$T_{\Phi, \Lambda}(h) = \sum_{j=1}^{\infty} \lambda_j (h, \phi_j) \phi_j, \tag{14}$$

**Theorem B.3** (Spectral Decomposition). *Suppose $H$ is a Hilbert space. A symmetric positive-definite Hilbert-Schmidt operator $T : \mathbb{H} \to \mathbb{H}$ admits the spectral decomposition equation 14 with orthonormal $\phi_j$ which are the eigenfunctions of $T$, i.e. $T(\phi_j) = \lambda_j \phi_j$. The $\phi_j$ can be extended to a basis by adding a complete orthonormal system in the orthogonal complement of the subspace spanned by the original $\phi_j$.*

*Remark* B.4. The $\phi_j$ in equation 14 can thus be assumed to form a basis, but some $\lambda_j$ may be zero.

From Horváth & Kokoszka (2012). Theorem proved in Gohberg (1990). Denote by $\mathcal{L}$ the space of bounded (continuous) linear operators on $\mathbb{H}$ with the norm

$$\|T\|_{\mathcal{L}} = \sup\{\|T(x)\| \mid \|x\| \le 1\}.$$

**Definition B.5** (Compact Operators). An operator $T \in \mathcal{L}$ is said to be compact if there exist two orthonormal bases $\{g_j\}$ and $\{f_j\}$, and a real sequence $\{\lambda_j\}$ converging to zero, such that

$$T(h) = \sum_{j=1}^{\infty} \lambda_j (h, g_j) f_j, \quad h \in \mathbb{H}, \tag{Compact}$$

The $\lambda_j$ may be assumed positive. The existence of representation equation Compact is equivalent to the condition: $T$ maps every bounded set into a compact set. Compact operators are also called completely continuous operators. Representation equation Compact is called the singular value decomposition.

**Definition B.6** (Hilbert-Schmidt Operators). A compact operator admitting representation equation Compact is said to be a Hilbert-Schmidt operator if $\sum_{j=1}^{\infty} \lambda_j^2 < \infty$. The space $\mathcal{S}$ of Hilbert-Schmidt operators is a separable Hilbert space with the scalar product

$$\langle T_1, T_2 \rangle_{\mathcal{S}} = \sum_{i=1}^{\infty} (T_1(f_i), T_2(f_i)), \tag{15}$$

where $\{f_i\}$ is an arbitrary orthonormal basis. Note the value of equation 15 is independent of the basis. The corresponding norm is

$$\|T\|_{\mathcal{S}}^2 = \sum_{j \ge 1} \lambda_j^2 \tag{HS}$$

One can show that

$$\|T\|_{\mathcal{L}} \le \|T\|_{\mathcal{S}}$$

**Definition B.7.** An operator $T \in \mathcal{L}$ is said to be symmetric if

$$\langle T(f), g \rangle = \langle f, T(g) \rangle, \quad f, g \in \mathbb{H},$$

and positive-definite if

$$\langle T(f), f \rangle \geq 0, \quad f \in \mathbb{H}.$$

(An operator with the last property is sometimes called positive semidefinite, and the term positive-definite is used when the inequality is strict.)

## C  DATA AUGMENTATION KERNEL PERSPECTIVE OF NON-CONTRASTIVE SSL

**Theorem C.1.** *Let $G(x)$ be the infinite Mercer features of the backward data augmentation covariance kernels, $k^{DAB}$. Let $F(x) = (f_1(x), f_2(x), \ldots, f_k(x))$ be the features given by minimizing the following data augmentation invariance loss*

$$L(F) = \sum_{i=1}^{N_k} \|T_M f_i - f_i\|_{L^2(\rho_X)}^2, \quad \text{subject to} \quad (f_i, f_j)_{\rho_X} = \delta_{ij} \tag{16}$$

*which includes the orthogonality constraint. Then, $V(F) \subset V(G)$, $V(F) \to V(G)$ as $N_k \to \infty$.*

The idea of the proof uses the fact that, as linear operators, $T_{k^{DAB}} = T_M^\top T_M$ and that $T_{k^{DAF}} = T_M T_M^\top$. Then we use spectral theory of compact operators, which is analogue of the Singular Value Decomposition in Hilbert Space, to show that eigenfunctions of $T_M^\top T_M$ operator are the same as those obtained from optimizing $L(F)$. A similar result can be obtained using $k^{DAF}$ and $T_M^\top$.

Note that $L(F)$ is the constrained optimization formulation of the BarlowTwins loss. Furthermore, $L(F)$ with the additional constraint that $(f_i, f_i) \geq \gamma \; \forall i \in \{1, 2 \ldots N_k\}$ is the constrained optimization formulation of the VICReg loss.

### C.1  PROOF OF THEOREM 3.2

We show we can factor the linear operator, leading to a practical algorithm. Here, we show that we can capture the backward data augmentation kernel with the forward data augmentation averaging operator

**Lemma C.2.** *Using the definitions above, and with $k$ in equation 8 given by $k^{DAB}$,*

$$T_k = T_M^\top T_M$$

*Proof.* First, define the non-negative definite bilinear form

$$B^{VAR}(f, g) = (T_M f, T_M g)_{\rho_X} \tag{17}$$

Given the backwards data augmentation covariance kernel, $k^{DAB}$, define

$$B^{DAB}(f, g) = (T_k f, g)_{\rho_X}$$

We claim, that

$$B^{VAR} = B^{DA,B} \tag{18}$$

This follows from the following calculation,

$$B^{DA,B}(f, g) = (T_k f, g)_{\rho_X} \tag{19}$$

$$= \mathbb{E}_x[T_k f(x), g(x)] = \mathbb{E}_x \mathbb{E}_z[k_{DA,B}(z, x) f(z) g(x)] \tag{20}$$

$$= \mathbb{E}_x \mathbb{E}_z \mathbb{E}_{x_0} \left[ \frac{p(x_0 \mid x)}{\rho(x_0)} \frac{p(x_0 \mid z)}{\rho(x_0)} f(z) g(x) \right] \tag{21}$$

$$= \mathbb{E}_{x_0} \left[ \sum_x \left( \frac{\rho(x) p(x_0 \mid x)}{\rho(x_0)} g(x) \right) \sum_z \left( \frac{\rho(z) p(x_0 \mid z)}{\rho(x_0)} f(z) \right) \right] \tag{22}$$

$$= \mathbb{E}_{x_0} \left[ \sum_x (p(x \mid x_0) g(x)) \sum_z (p(z \mid x_0) f(z)) \right] \quad \text{[Using Bayes' rule]} \tag{23}$$

$$= \mathbb{E}_{x_0} [T_M f(x_0) T_M g(x_0)] = (T_M f, T_M g)_{\rho_X} = B^{VAR}(f, g) \tag{24}$$

$\square$

For implementations, it is more natural to consider *invariance* to data augmentations.

**Theorem C.3** (equivalent eigenfunctions). *Assume that $T_M$ is a compact operator. Define the invariance bilinear form*

$$B^{INV}(f,g) = (T_M f - f, T_M g - g) \tag{25}$$

*Then $B^{INV}$, $B^{VAR}$ share the same set of eigenfunctions. Moreover, these are the same as the eigenfunctions of $B^{DA,B}$. In particular, for any eigenfunction $f_j$ of $B^{VAR}$, with eigenvalue $\lambda_j$, then $f_j$ is also and eigenfunction of $B^{INV}$, with the corresponding eigenvalue given by $(\sqrt{\lambda_j} - 1)^2$.*

*Proof.* Define $T_{MM}$ by,

$$T_{MM} f = T_M^\top T_M f \tag{26}$$

Define

$$T_{MS} = (T_M - I)^\top (T_M - I) \tag{27}$$

Note, by the assumption of compactness, $T_M$ has the Singular Value Decomposition, (see the Hilbert Space section for equation SVD),

$$T_M(h) = \sum_{j=1}^{\infty} \lambda_j (h, g_j) f_j \tag{SVD}$$

Let $f_j$ be any right eigenvector of $T_M$, with eigenvalue $\mu_j$. Then $f_j$ is also a right eigenvector $T_M - I$, with eigenvalue $\mu_j - 1$. So we see that $T_{MM}$ has $f_j$ as an eigenvector, with eigenvalue $\lambda_j = \mu_j^2$ and $T_{MS}$ has $f_j$ as an eigenvector, with eigenvalue $(\sqrt{\lambda_j} - 1)^2$. Finally, the fact that there are no other eigenfunctions also follows from equation SVD.

The final part follows from the previous lemma. □

## D  MULTI-AUGMENTATION LEARNING

### D.1  AUGMENTATION GRAPH

We use the population augmentation graph formulation introduced in HaoChen et al. (2021). Briefly, we define a graph $\mathcal{G}(\mathcal{X}, \mathcal{W})$, where the vertex set $\mathcal{X}$ comprises of all augmentations from the dataset (could be infinite when continuous augmentation functions are used) and $\mathcal{W}$ denotes the adjacency matrix with edge weights as defined below:

$$w_{xx'} := \mathbb{E}_{\bar{x} \sim \mathcal{P}_{\bar{X}}} \left[ \mathcal{A}(x|\bar{x}) \mathcal{A}(x'|\bar{x}) \right] \tag{28}$$

, i.e. the joint probability of generating 'patches' $x, x'$ from the same image $\bar{x}$. Here $\mathcal{A}$ defines the set of augmentation functions used in the SSL pipeline. It is worth noting that the magnitude of $w_{xx'}$ captures the relative similarity between $x$ and $x'$. A higher value of $w_{xx'}$ indicates that it is more likely that both patches came from the same image, and thereby are more similar. The marginal likelihood of each patch $x$ can also be derived from this formulation:

$$w_x = \mathbb{E}_{x' \sim \mathcal{X}} \left[ w_{xx'} \right] \tag{29}$$

### D.2  CONTRASTIVE AND NON-CONTRASTIVE LOSSES SUFFER FROM THE SAME ISSUES

We will now show that the proposal of using multiple patches for the $\mathcal{L}_{invariance}$ is pertinent to both the contrastive and non-contrastive SSL. Following HaoChen et al. (2021), we use the spectral contrastive loss formulation and incorporate the augmentation graph relations:

$$\mathcal{L}_c = -\mathbb{E}_{x,x^+} \left[ f(x)^T f(x^+) \right] + \beta \mathbb{E}_{x,x'} \left[ \left( f(x)^T f(x') \right)^2 \right]$$

$$\mathcal{L}_c \propto \|ZZ^T - D^{-\frac{1}{2}} \mathcal{W} D^{-\frac{1}{2}}\|_F^2 = \|ZZ^T - \bar{\mathcal{W}}\|_F^2 \tag{30}$$

where $z := \sqrt{w_x} f(x)$, $D$ is a $N \times N$ diagonal matrix with entries $\{w_x\}$ and $\bar{\mathcal{W}} = D^{-\frac{1}{2}} \mathcal{W} D^{-\frac{1}{2}}$.

We extend the duality results between contrastive and non-contrastive SSL loss, established by Garrido et al. (2022), to demonstrate how eq. (30) can be decomposed into the invariance and collapse-preventing loss terms.

$$\|ZZ^T - \bar{\mathcal{W}}\|_F^2 = \|Z^T Z - I_d\|_F^2 + 2Tr\left[Z^T(I_N - \bar{\mathcal{W}})Z\right] + \kappa \tag{31}$$

$$= \|Z^T Z - I_d\|_F^2 + 2\sum_i \sum_x (1 - \bar{w}_x)z_i^2 - 2\sum_i \sum_{x,x'} \bar{w}_{xx'} z_i z_i' + \kappa \tag{32}$$

where $\kappa$ is some constant independent of $Z$. The first term in eq. (31) is the covariance regularization term in non-contrastive losses like BarlowTwins (implicit) or VIC-Reg (explicit), and the second term in eq. (32) is the variance regularization. Simplifying the third term in eq. (32) gives us:

$$\sum_i \sum_{x,x'} \bar{w}_{xx'} z_i z_i' = \sum_i \sum_{x,x'} w_{xx'} f(x)_i f(x')_i = \sum_i \sum_{x,x'} \mathbb{E}_{\bar{x}\sim\mathcal{P}_{\bar{X}}} \left[\mathcal{A}(x|\bar{x})\mathcal{A}(x'|\bar{x})f(x)_i f(x')_i\right]$$

$$= \sum_i \mathbb{E}_{\bar{x}\sim\mathcal{P}_{\bar{X}}} \left[\sum_x \mathcal{A}(x|\bar{x})(f(x)_i \overline{f(x)}_i - f(x)_i^2)\right]$$

$$= \mathbb{E}_{\bar{x}\sim\mathcal{P}_{\bar{X}}} \left[\sum_x \mathcal{A}(x|\bar{x})\left(f(x)^T \overline{f(x)} - \|f(x)\|^2\right)\right] \tag{33}$$

This term encourages $f(x)$ to be similar to $\overline{f(x)}$, i.e. the mean representation across all augmentations of $\bar{x}$, thereby requiring to "sufficiently" sample $A(.|\bar{x})$. Given that both the contrastive and non-contrastive losses rely on learning invariance properties from data augmentations, we believe that our multi-patch proposal would improve the probability density estimation of $A(.|\bar{x})$ and yield better performance with few training epochs.

### D.3 Explaining training dynamics in low patch sampling regime

We now turn to a simple form of the augmentation graph to understand how using low number of augmentations affects the evolution of $ZZ^T$. Minimizing eq. (30) implies that the spectral decomposition of $Z$ would align with the top eigenvectors (and values) of $\overline{\mathcal{W}}$. We will demonstrate that in the low sampling regime (using few augmentations), the eigenvectors of the sampled augmentation graph $\tilde{\mathcal{W}}$ *may not* align with those of $\overline{\mathcal{W}}$.

**Augmentation graph setup.** We define an augmentation graph with only two instances from two different classes, similar to the one presented in (Shen et al., 2022). Let us denote the four instances as $\bar{x}_i$ for $i \in 1, 2, 3, 4$, where $\bar{x}_1, \bar{x}_2$ belong to class 1 (i.e. $y_1, y_2 = 1$) and $\bar{x}_3, \bar{x}_4$ belong to class 2 (i.e. $y_3, y_4 = 4$). Let us further assume that $\bar{x}_1, \bar{x}_3$ have the highest pixel-level similarity among $(\bar{x}_1, \bar{x}_i)\forall i \in 2, 3, 4$, thereby making it more likely to have similar patches (see Figure 2 for illustration). We denote this relationship among input examples using $\mathcal{G}$ to indicate (pixel-wise) global similarity groups. So, $\mathcal{G}_1, \mathcal{G}_3 = 1$ and $\mathcal{G}_2, \mathcal{G}_4 = 2$. We can use the following probabilistic formulation to model our augmentation functions (see Figure 2B):

$$A(x_j|\bar{x}_i) = \begin{cases} \rho' & \text{if } j = i \\ \mu' & \text{if } j \neq i \text{ and } y_j = y_i \text{ and } \mathcal{G}_j \neq \mathcal{G}_i \\ \nu' & \text{if } j \neq i \text{ and } y_j \neq y_i \text{ and } \mathcal{G}_j = \mathcal{G}_i \\ \delta' & \text{if } j \neq i \text{ and } y_j \neq y_i \text{ and } \mathcal{G}_j \neq \mathcal{G}_i \end{cases} \tag{34}$$

In our setting, $\rho' + \mu' + \nu' + \delta' = 1$. The adjacency matrix of our augmentation graph (as shown in Figure 2C) is as follows:

$$\overline{\mathcal{W}} = \begin{bmatrix} \rho & \mu & \nu & \delta \\ \mu & \rho & \delta & \nu \\ \nu & \delta & \rho & \mu \\ \delta & \nu & \mu & \rho \end{bmatrix} \tag{35}$$

We defer the relations between $\rho', \mu', \nu'\delta'$ and $\rho, \mu, \nu, \delta$ to the appendix. The eigenvalues of this matrix are: $(\rho + \mu + \nu + \delta, \rho + \mu - \nu - \delta, \rho - \mu + \nu - \delta, \rho - \mu - \nu + \delta)$. Corresponding

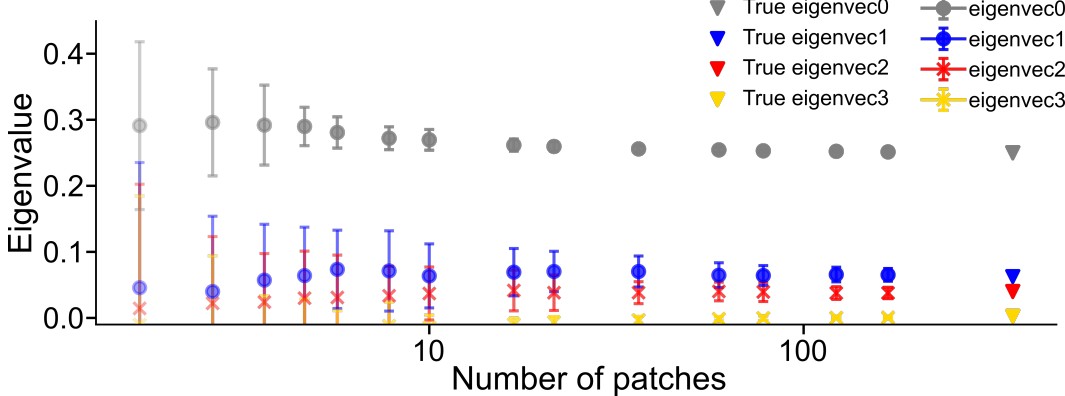

Figure 10: Empirical verification of the subsampling Ansatz.

eigenvectors are along $[1,1,1,1]^T$, $[1,1,-1,-1]^T$. $[1,-1,1,-1]^T$, $[1,-1,-1,1]^T$. Assuming that the augmentation functions induce semantically-relevant invariance properties that are relevant for identifying $y_i$ from $f(x_i)$, we can say that $\rho' > max\{\mu', \nu'\}$ and $min\{\nu', \mu'\} > \delta'$. When we have sufficiently sampled the augmentations, any SSL loss will learn $Z$ such that its singular values are span the top eigenvectors of the augmentation graph, and the eigenspectrum of $ZZ^T$ would simply be the above eigenvalues. In practical settings, the augmentation graph would have significantly higher dimension that the feature/embedding dimension [1]. Therefore, singular vectors of $Z$ would span the top eigenvectors of $\overline{\mathcal{W}}$ and the smaller eigenmodes are not learned. When we have accurately sampled the augmentation graph, $\mu > \nu$ and therefore, the class-information preserving information is preferred over pixel-level preserving information during learning. But what happens when we *do not sufficiently sample the augmentation space?*

**Ansatz.** Based on our empirical experience, we define *an ansatz* pertaining to the eigenvalues of a sampled augmentation graph and validate it in tractable toy settings, such as the one described above. Specifically, we claim that when the augmentation space is not sufficiently sampled, $\{|\mu - \nu|, \delta\} \to 0$. In other words, we claim that when only few augmentations per example are used, it is more likely to have an equal empirical likelihood for augmentations that preserve (pixel-level) global information and class/context information. Moreover, it is very unlikely to have augmentations that change both the class and global information. This is demonstrated in Figure 10.

**Consequences of the *Ansatz*.** When only a few augmentations are sampled, learning can suppress the class information at the cost of preserving the pixel-level information, thereby leading to an increased smoothness in the learned feature space.

## E  IMPLEMENTATION DETAILS

**Image Classification Datasets** Across all experiments, our settings mainly follow Chen et al. (2022). In particular, we run Table 3a summarizes our pretraining settings on Cifar-10 Krizhevsky & Hinton (2009), and STL-10 Coates et al. (2011). In Table 3b, we outline the corresponding linear evaluation settings for Resnet-50. Note that we add a linear classifier layer to the encoder's features and discard the projection layers for evaluation.

### E.1  EMPIRICAL RESULTS ON TRANSFER LEARNING

In this section, we present extended version of results presented in Figure 4, Figure 5 but pretraining on CIFAR-10 (or STL-10) and evaluating on STL-10 (or CIFAR-10). These results, coupled with the ones in Figure 4 Figure 5, present a strong case for the advantage of using the proposed multi-augmentation loss for better convergence as well as downstream accuracy.

---

[1]Contrastive algorithms use a large batch size, thereby optimizing a high-dimensional $ZZ^T$ whereas non-contrastive algorithms use a large embedding dimension, thereby optimizing a high-dimensional $Z^T Z$.

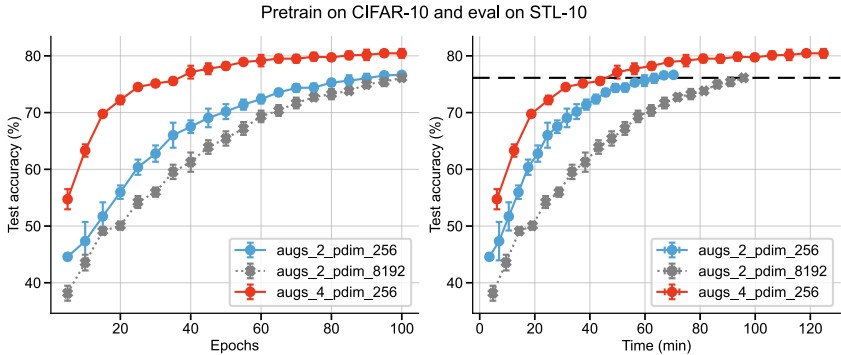

Figure 11: BarlowTwins pretraining on CIFAR-10, linear evaluation on STL-10 labelled set.

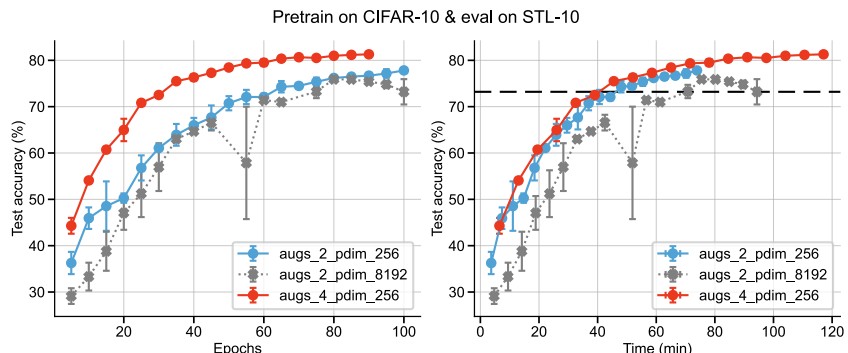

Figure 12: VICReg pretraining on CIFAR-10, linear evaluation on STL-10 labelled set.

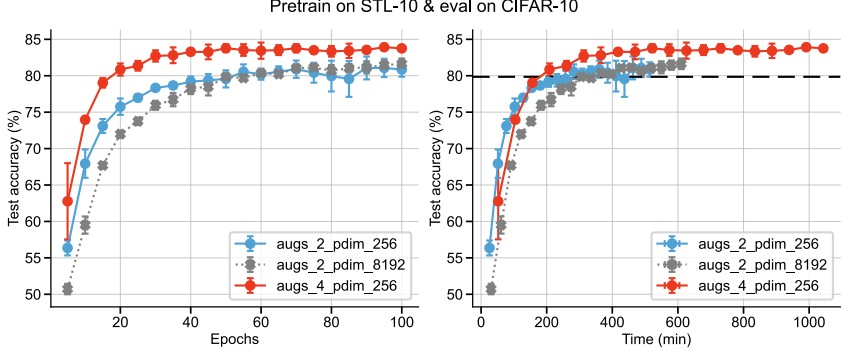

Figure 13: BarlowTwins pretraining on STL-10, linear evaluation on CIFAR-10 labelled set.

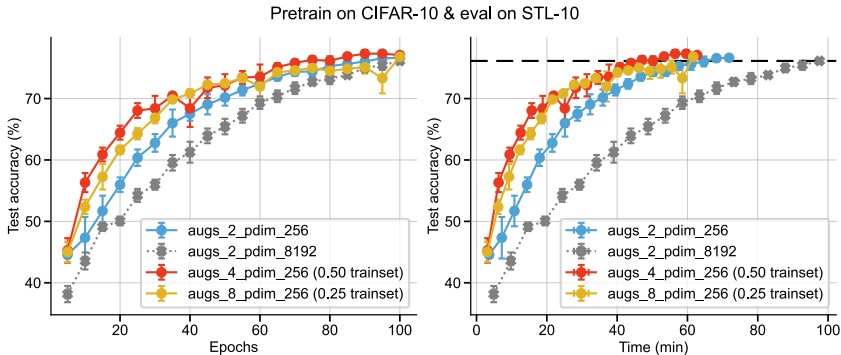

Figure 14: BarlowTwins pretraining on fraction of CIFAR-10 trainset, linear evaluation on STL-10 labelled set.

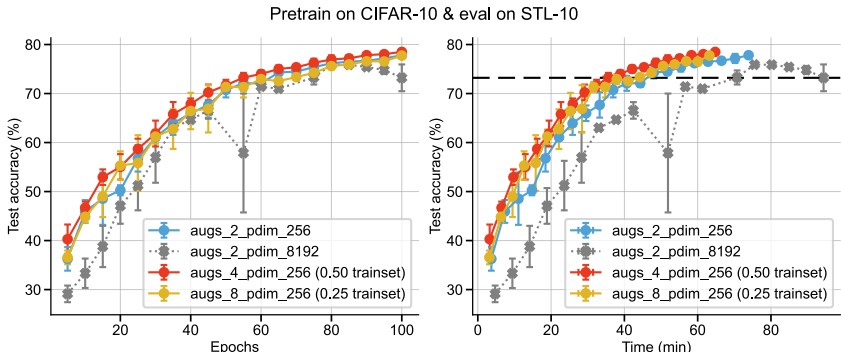

Figure 15: VICReg loss pretraining on fraction of CIFAR-10 trainset, linear evaluation on STL-10 labelled set.

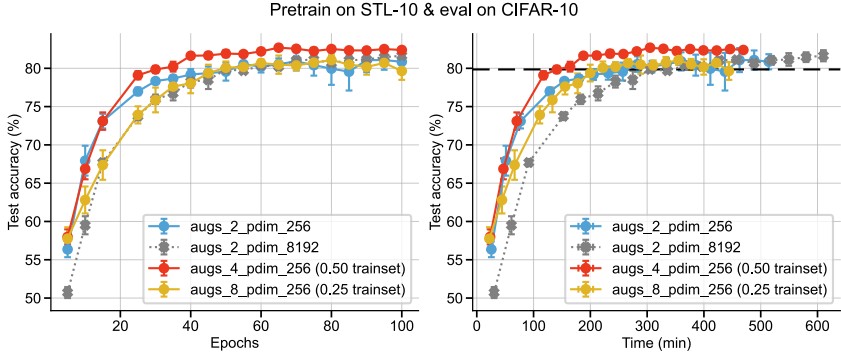

Figure 16: BarlowTwins loss pretraining on fraction of STL-10 unlabelled set, linear evaluation on CIFAR-10 train set.

| config | value |
|--------|-------|
| optimizer | Adam |
| learning rate | 1e-3 |
| batch size | 128 |
| epochs | 100 (CIFAR10), 50 (STL10) |
| weight-decay | 1e-6 |

(a) Pretraining

| config | value |
|--------|-------|
| optimizer | Adam |
| learning rate | 1e-3 |
| batch size | 512 |
| epochs | 200 |
| weight-decay | 1e-6 |
| test-patches | 16 |

(b) Linear Evaluation

Table 3: Experiment Protocol for comparing SSL algorithms

