# OpenReview forum: "Addressing Sample Inefficiency in Multi-View Representation Learning"
_ICLR.cc/2024/Conference — Submitted to ICLR 2024_

### Official Review · Reviewer_QPCx · 2023-10-25

**Soundness:** 2 fair
**Presentation:** 3 good
**Contribution:** 2 fair
**Rating:** 6
**Confidence:** 3

**Summary:**

This paper analyzed two self-supervised contrastive methods including Barlow Twins and VICReg. Based on these loss functions, they reveal that the feature's orthogonality has more impact than the projection head's dimensionality. Next, they find that using more data augmentations for each image can improve the learned representations and reduce the training dataset size while not sacrificing accuracy. Experiments are conducted in small datasets CIFAR-10 and STL-10.

**Strengths:**

+ It provides a new perspective on SSL with non-contrastive approaches.
+ Self-supervised learning with a strategy that reduces the data needed to learn good representations may benefit the community.
+ The writing is clear

**Weaknesses:**

There are several concerns that have been raised:
+ The finding that more data augmentations can help learn better representations is well-known, many works already show that multi-crop can help contrastive learning achieve better performance (DINO [1], SWAV [2], MSF [3], iBOT, etc ...)

+ Experiments are insufficient to demonstrate their effectiveness where only small-scale datasets (in terms of both resolution and dataset size) are conducted. I recommend verifying and evaluating the method on more challenging datasets such as ImageNet 224x224, which is a benchmark in this field.

+ Several Non-contrastive methods such as BYOL and SimSiam or DINO have not involved any negative samples but they are not adequately compared.

+ The experimental setting is also not sufficient since it only trained for 100 epochs on CIFAR-10 and 50 epochs on STL-10, some methods with more augmentations can have a fast convergence, but for longer training that might be diminished. This setting is not practical where most SSL methods are not converged (as shown in the SimSiam paper). Please see [4] for reference where all SSL methods are conducted at least 1000 epochs.

+ Lack of comparisons with other SSL methods would weaken the impact of the paper.

+ The paper claims that more augmentations would help the representation learning quality but some figures show that 8 augmentations perform worse than 4 augmentations, this may be contradicting to that claim.


[1] Emerging Properties in Self-Supervised Vision Transformers, ICCV 2021 \
[2] Unsupervised Learning of Visual Features by Contrasting Cluster Assignments, NeurIPS 2020 \
[3] Mean Shift for Self-Supervised Learning, ICCV 2021 \
[4] solo-learn: A Library of Self-supervised Methods for Visual Representation Learning, JMLR 2022

**Questions:**

See weaknesses

---

> ### Author Response · Authors · 2023-11-16
> **Official response to Reviewer QPCx [1/2]**
>
> We thank the reviewer for their detailed feedback and highlighting the core contributions of our work. We are also grateful to the reviewer for appreciating that “writing is clear,” and our insights suggest a “strategy that reduces the data needed to learn good representations” that “may benefit the community.” We would like to clarify certain aspects of our work, especially our focus on the theoretical foundations of correlation-based non-contrastive self-supervised learning (NC-SSL) methods and the specific contributions of our research.
>
> 1. ### Emphasis on Theoretical Contributions:
>
> We would like to clarify that we agree with the reviewer that multi-view or data augmentation for non-contrastive SSL are well-established tools in the literature (see Sec. 5, where we cover related work, including multi-view SSL) but do not claim this to be our core contribution. Instead, we aim to offer a principled and theoretically grounded understanding of the features learned by pretraining with correlation-based non-contrastive SSL frameworks, specifically Barlow Twins and VICReg. In particular, we show that pretraining with these loss functions recovers features that span the top eigenfunctions of the augmentation-defined data kernel. Building on these insights, we present practical recommendations to make SSL training more compute and sample efficient. Therefore, we offer a deeper understanding of these algorithms and why specific useful heuristics (e.g., using multiple augmentations) are helpful in good feature learning.
>
> 2. ### Longer Pretraining:
>
> We thank the reviewer for sharing solo-learn [1] as a reference for several SSL algorithms pretrained for 1000 epochs. However, we would like to highlight that while solo-learn pretrains for 1000 epochs on CIFAR-10, they consider only Resnet-18 architectures that are computationally much cheaper than the Resnet-50 used in our experiments, making it difficult to run 1000-epoch pretraining on our compute budget. Nevertheless, we worked hard over the last few days to incorporate the reviewer’s feedback. We have updated Fig. 4 & 5 with 100-epoch pretraining on STL-10 and 400-epoch pretraining on CIFAR-10 (see Appendix A). Note that our core claims still hold, i.e., using multiple augmentations leads to good downstream task performance earlier in training.
>
>
> | **Algorithm**                          | **Best Accuracy (%)** | **Epoch of Best Accuracy** |
> |:--------------------------------------:|:---------------------:|:-----------------------------:|
> | Barlow-Twins (2-augs) w/ pdim=256      $\hspace{1cm}$ | 92.04 ± 0.16       $\hspace{1.5cm}$   | 400                      $\hspace{1cm}$     |
> | Barlow-Twins (4-augs) w/ pdim=256      $\hspace{1cm}$ | 92.39 ± 0.17        $\hspace{1.5cm}$  | 340                      $\hspace{1cm}$     |
> | Barlow-Twins (8-augs) w/ pdim=256      $\hspace{1cm}$ | 92.64 ± 0.10       $\hspace{1.5cm}$  | 140                      $\hspace{1cm}$     |
>
>
>
> 3. ### Addressing Misinterpretations of 8 Augmentations Results:
>
> We would like to clarify the misunderstanding regarding the performance of 8 augmentations.
> - First, Fig. 4 demonstrates the utility of multiple augmentations in improving convergence, i.e., learning better features earlier in training using the full dataset. We have added results for 8 augmentations while pretraining with BarlowTwins loss on CIFAR-10 to demonstrate that **increasing the number of augmentations leads to better downstream performance earlier in training** (see Appendix A), in line with our claims.
> - On the other hand, Fig. 5 inspects the limits of SSL feature learning with fewer unique unlabeled samples. To this end, we vary the dataset size such that the total number of samples (after augmentations) in each epoch is the same, i.e., we compare training on 100% dataset with 2 augmentations, 50% dataset with 4 augmentations, and 25% dataset with 8 augmentations. The reviewer is correct in pointing out that training with 25% of the dataset with 8 augmentations can slightly hurt downstream performance compared to training with 50% dataset with 4 augmentations. We note this is not a discrepancy between our theory and empirical results. Instead, this indicates that while **multiple augmentations can recover competitive performance with fewer samples**, there is a **limit to their utility in the extremely low data regime**. To further highlight this tradeoff between downstream task performance and the number of unique unlabeled samples, we vary the number of augmentations and dataset size to develop an improved Pareto frontier, as shown in Fig. 6.

---

> ### Author Response · Authors · 2023-11-16
> **Official response to Reviewer QPCx [2/2]**
>
> 4. ### On additional large-scale experiments and baselines:
>
> a. **Theoretical Corroboration**: Our core contributions include theoretical results that explain why heuristics like multiple augmentations and high-dimensional projectors help with self-supervised pretraining. Our experiments demonstrate that these insights are practically valuable and translate to improvement in performance, convergence, and sample efficiency. In this context, CIFAR-10 and STL-10, while smaller in scale, are nonetheless relevant and provide a robust platform for demonstrating the efficacy of our method. Our results align well with theoretical expectations, indicating that our findings are not merely incidental but are rooted in solid theoretical understanding, and we expect this to translate to other settings, including experiments with larger datasets (such as Imagenet).
>
> b. **Adding Imagenet Experiments**: While we acknowledge that ImageNet is a benchmark in representation learning, we must consider the practical constraints of computational resources.
> Given our current computing budgets, it is difficult to report results (with proper hyperparameter tuning) on ImageNet. However, we are trying hard to incorporate these results within the rebuttal deadline, and if our compute budget permits, we will follow-up the reviewer’s concern with results on ImageNet.
>
> c. **Comparisons with Other SSL Methods**: Regarding the comparisons with other Self-Supervised Learning (SSL) methods like BYOL, SimSiam, and DINO, while we agree that a comprehensive comparison is valuable, we believe these comparisons are outside the scope of this work. In particular, we seek to establish theoretical foundations for correlation-based non-contrastive SSL. We aim to extend our theoretical framework to other families of non-contrastive SSL, with an eye toward practical recommendations to improve these methods.
>
>
> In conclusion, our paper aims to strengthen the theoretical underpinnings of correlation-based SSL methods, providing novel and fundamental insights into the field. We appreciate the opportunity to clarify these aspects and believe our work contributes significantly to a deeper understanding of SSL. We hope this clarification addresses the concerns raised and demonstrates the value and rigor of our research.

---

> > ### Comment · Reviewer_QPCx · 2023-11-22
> >
> > Thanks for the rebuttal, I recognize it addressed some of my concerns. I am happy to increase the score. I hope the reviews are helpful in strengthening their paper.

---

> > > ### Author Response · Authors · 2023-11-22
> > > **Thank you note**
> > >
> > > Thank you for revisiting our paper and adjusting your score. Your feedback has indeed improved our work, and we're thrilled to see the acceptance rating. Once again, thank you for time and valuable input.

---

### Official Review · Reviewer_ArGT · 2023-10-25

**Soundness:** 2 fair
**Presentation:** 2 fair
**Contribution:** 1 poor
**Rating:** 3
**Confidence:** 4

**Summary:**

This submission follows a recent line of work and studies self-supervised representation learning from a kernel perspective. It shows that non-contrastive learning methods such as Barlow Twins and VICReg can find the eigenfunctions of the integral operator of the augmentation kernel (positive-pair kernel), and then claims that a low-dimensional representation is sufficient, and using more diverse augmentations can improve pretraining.

**Strengths:**

The manuscript is easy to read in general.

**Weaknesses:**

As someone closely following the literature, I feel that a large part of this submission has already been covered by two prior work: [1, 2], and I don't really find anything particularly new in this submission. ([1] was in ICLR last year and [2] was on arXiv in June.) Moreover, the writing of this submission is quite confusing at times. Especially, the mathematical part is not very rigorous and needs a lot of improvement. Thus, I recommend rejecting this submission. However, the subject matter of this submission is definitely very interesting, and I encourage the authors to dive deeper into this field. I also recommend the authors to read [3, 4], which the authors might have overlooked. In particular, [4] comes from the same group as VICReg, and covers most results about VICReg in this submission.

My detailed comments are the following:
### 1. On the nature of non-contrastive learning, and more generally, augmentation-based self-supervised learning
This submission shows that non-contrastive learning is essentially approximating the kernel of the augmentation graph (called the positive-pair kernel in [1]). This is a known result. For spectral contrastive learning, this has been shown by [1]; For more general augmentation-based self-supervised learning, this has been shown by [2]. In particular, Appendix C of [2] shows that Barlow Twins and VICReg are minimized when the representation recovers the linear span of the top-d eigenfunctions of $T_k$, the integral operator of the augmentation kernel. This result is stronger than Theorem 3.1 in this submission.

Moreover, the nature of other SSL algorithms such as MAE is not an "open problem" (page 9), as it has already been addressed by [2].

### 2. On the mathematical writing of this work
The writing of this work is quite confusing at times, especially in the mathematical part:
- In Theorem 3.1, what does $V(F) \rightarrow V(G)$ mean exactly? In functional analysis, "a sequence of subspaces converges" usually means that the sequence of their projection operators converge under some norm, such as the operator norm or the Hilbert-Schmidt norm. I guess the authors want to mean convergence under the HS norm.
- The definition of $k^{DAB}$ and $T_M$ seems strange, and I guess that's why the authors cannot prove that V(F) is the linear span of the top-d eigenfunctions. The problem is that $T_Mf(x) = \int f(x_0) p(x_0|x) dp(x_0)$, so there are two $p(x_0)$ on the numerator. A better definition could be $T_M f(x) = \int f(x_0) dp(x_0|x)$. I suggest the authors read Section 2.2 of [2], and replace $k^ {DAF}, k^{DAB}$ with the $K_ A, K_ X$ defined there.
- Sections 3.2 and 3.3 are not "corollaries". Maybe "insight" is a better term.
- In Section 3.2 "Low-dimensional projectors are sufficient", what is the definition of "sufficient"? Does it mean that the ground truth target function could be reconstructed? Or does it mean that the error could be arbitrarily small? The effect of $d$, the representation dimension, has been studied in [2, 3]. Specifically, [2] showed that a larger $d$ leads to a smaller approximation error but a larger estimation error, so there is a trade-off and no $d$ is perfect or "sufficient".
- In Section 3.3 "Multiple augmentations improve optimization", what is the definition of "improve optimization"? Does it mean faster convergence rate? Or does it mean more stable optimization, or perhaps lower sharpness? I don't think this section is talking about optimization at all.
- In Section 3.3, the authors wrote "using only two augmentations per sample yields a noisy estimate of $T_M$", which seems to suggest that using more augmentations leads to better estimate of $T_M$. The problem is that the definition of $T_M$ depends on $M$, the augmentation. So if there are more augmentations, then $T_M$ would not be the same $T_M$. Thus, I find this statement really confusing.
- The title of this submission is "addressing sample inefficiency in ...", but what is "sample efficiency"? Section 4.3 seems to be the only section in the main body addressing sample efficiency, and this section uses an experiment to show that if more, diverse augmentations are used, then fewer samples are required. I guess the authors are trying to express that using more diverse augmentations can lead to lower sample complexity, which is actually a known result. Specifically, [2] showed that stronger augmentations lead to lower sample complexity, and here "stronger" includes "more, diverse" augmentations.

Finally, in the summary the authors position this submission as providing "a fresh theoretical analysis", but (a) most results are known results and thus are not really fresh, and (b) the only theorem in the submission is Theorem 3.1, and I feel that this work is more on the empirical or heuristic side. Regarding the experiments, they are kind of interesting, but I have seen similar experiments in prior work. Foundation models and representation learning are such popular topics these days, so I think the authors really ought to do a much more thorough literature review, before claiming anything to be fresh or new.


[1] Johnson et al., Contrastive Learning Can Find an Optimal Basis for Approximately View-invariant Functions, ICLR 2023.
[2] Zhai et al., Understanding Augmentation-based Self-supervised Representation Learning via RKHS Approximation and Regression, arXiv:2306.00788.
[3] Saunshi et al., Understanding Contrastive Learning Requires Incorporating Inductive Biases, ICML 2022.
[4] Cabannes et al., The SSL Interplay: Augmentations, Inductive Bias, and Generalization, ICML 2023.

**Questions:**

See my detailed comments above. Overall, I suggest the authors carry out a more thorough literature review.

---

> ### Author Response · Authors · 2023-11-17
> **Official response to Reviewer ArGT**
>
> We acknowledge the reviewer’s engagement with our paper. While some points raised and references cited by the reviewer are interesting and contribute to our overall understanding of non-contrastive SSL, we would like to highlight several aspects of the review that necessitate further discussion.
>
> 1. ### Note on ICLR review guidelines:
>
> Most of the reviewer’s criticism of our theoretical contribution relies on [2] Zhai et al paper available on arXiv in June 2023. However, this contradicts the ICLR guidelines: “consider papers contemporaneous if they are published (available in online proceedings) within the last four months” (https://iclr.cc/Conferences/2024/ReviewerGuide#FAQ). The same goes for [4] Cabannes et al. 2023, published at ICML 2023 on July 03, 2023 \[[proceedings](https://proceedings.mlr.press/v202/).\]
>
> 2. ### Feedback on theoretical typos:
>
> We thank the reviewer for pointing out some of the typos in the theory sections of our paper. Indeed, as the reviewer pointed out, we have a typo in the definition of $T_M$, which should follow the definition in the text (see the line below). We will update the writing in the theory section of our manuscript, per the reviewer’s suggestions, but we would like to point out that these do not change the core takeaway of our work. Moreover, we feel that the reviewer’s feedback either focuses on minor points that detract from a holistic evaluation of our work or lacks specific constructive feedback on improving our work. We would therefore request the reviewer to kindly offer more specific points or concerns they may have regarding the theoretical arguments presented in our work.
>
> 3. ### Misinterpretation of the scope of our work:
> The reviewer cites several references ([1], [3], [4]) as superior or more definitive than our findings. However, these references predominantly offer theoretical and potential insights without concrete experimental validation. Furthermore, [1] & [3] specifically deal with contrastive learning frameworks (as the reviewer also notes). In the related work section, we mention how our work builds on insights from theoretical advances in contrastive SSL and cite [1]. While we agree with the reviewer that these works present useful theoretical insights, we strongly disagree that these works are “superior” to our work. On the contrary, we believe that our work presents complementary viewpoints, thereby improving the overall quality of scientific know-how in this research direction. For example, Section 6.2 in [4] mentions, “However, when the total amount $m \times n$ of pre-processed data is held fixed, it is generally better to process many inputs with two views $m = 2$, rather than a few inputs with many augmentations.” Our results (Fig. 5 & 6) show that this statement is not generally true in practical settings and that a tradeoff exists where multiple augmentations can sometimes yield better performance with fewer unlabeled samples. Therefore, it is indeed possible to use multiple augmentations to improve the sample efficiency of SSL algorithms, contrary to the recommendation provided by [4]. Unfortunately, the reviewer has currently overlooked these empirical contributions of our work.
>
> We would like to reiterate that our work is a timely research (evidenced by concurrent submissions addressing similar questions) that marries theoretical insights with practical recommendations, thereby bridging the gap between SSL theory and practice. Specifically, we focus on non-contrastive SSL algorithms like BarlowTwins and VICReg. Based on the reviewer’s comments, we agree that it would be interesting to adopt a similar approach as our work and extend existing theoretical literature to present practical recommendations that apply to more general SSL algorithms. Therefore, our work, in addition to the concurrent works (mentioned by the reviewer), contributes to a better understanding of SSL algorithms and paves the way for more compute and sample-efficient SSL frameworks. However, that is outside the scope of our current paper.
>
> In conclusion, we urge the reviewer to reassess our paper, considering its scope, content, and contributions rather than comparing/contrasting it against external publications, especially those not aligned with ICLR guidelines. We believe our work represents a significant step forward in SSL research by presenting a holistic perspective from theoretical insights to practical recommendations, thereby drawing the attention of SSL practitioners to recent theoretical developments.

---

> > ### Comment · Reviewer_ArGT · 2023-11-17
> > **Reply to Author Response**
> >
> > I thank the authors for their response. I regret to see that the authors completely misunderstood my review, and wrote an emotional response that addresses almost none of my comments and, to be frank, is not very professional.
> >
> > I would like the authors to know that the sole purpose of my review is to help the authors know more about this field and thereby improve their work. I took a very long time to write my review, and I believe that I provided in that review very specific suggestions on how to improve this work, including related papers the authors might have overlooked, and specific points on the unrigorous mathematical writing. Sure I gave a “reject” rating which the authors might find annoying, but this is my honest opinion which I stand by. And I am quite confident of my judgment, because like I said in my review, I follow this literature very closely, and I believe that anyone having read my review would be convinced by this point.
> >
> > I would like to respond to the authors’ response as follows:
> >
> > ### 1. Contemporary work?
> > I am well aware of the ICLR review guidelines. In fact, had this paper proved results that are similar to Zhai et al. (2023) or Cabannes et al. (2023), then I would view them as contemporary work and rated “accept”. My point is that the theoretical results in this work are much weaker than Zhai et al. (2023) and Cabannes et al. (2023). For example, this work only shows that $V(F) \rightarrow V(G)$ when $d \rightarrow \infty$ (which by the way is not rigorous), while both Zhai et al. and Cabannes et al. showed that $V(F)$ consists of the top-d eigenfunctions. The empirical results are nothing new either. So provided that there are already two papers that contain much stronger results than those in this work, it is really hard for me to rate “accept”.
> >
> > Besides, even if Zhai et al. (2023) and Cabannes et al. (2023) did not exist, I would not give “accept” to this submission, because like I said in my review, the mathematical writing is too unrigorous and confusing, the results are not sound, and the claims are misleading.
> >
> > ### 2. Theoretical “typos”?
> > I do not understand why the authors are calling the problems I pointed out as “typos”. Please keep in mind that ICLR is a venue for serious mathematical and scientific research. Ambiguous conclusions such as $V(F) \rightarrow V(G)$ in Theorem 3.1 are not rigorous and, to be clearer, are incorrect. What could “a series of finite-dimensional spaces converge to an infinite-dimensional space” possibly mean? This is far more than just “typos”. This is an incorrect statement, and must be edited before this work could be accepted by any scientific venue, let alone a top venue like ICLR.
> >
> > I also raised other questions in my review. For example, what is meant by “sufficient” in Section 3.2, and why is Section 3.3 talking about optimization. It really baffles me why the authors would call them “minor points”, not “constructive”, and just “typos”. The authors also wrote in the response that this work “marries theoretical insights with practical recommendation”. I would very frankly say that I wouldn’t recommend any practitioners to follow the recommendations in this work, because they come from unsound, unrigorous theoretical analysis and can be misleading.
> >
> > ### 3. Scope of this work
> > My understanding is that the authors themselves positioned this work as a theoretical work, because in their conclusion, they wrote “our work presents a fresh theoretical analysis that sheds light on the implicit bias of non-contrastive SSL algorithms”. That is why my review mainly focused on the theoretical part. Had the authors positioned this work as an empirical work, I would have written my review differently, mainly focusing on the experiments. But even in that case, provided that there are so many empirical papers on contrastive/non-contrastive learning in 2022 and 2023, I really cannot see what is new in the experiments of this work.
> >
> > **Summary**: Frankly speaking, I am very disappointed to see that the authors misinterpret my good will of helping them improve their work. I know that receiving a “reject” is annoying, but this is a part of doing research, and in fact a very necessary and important part. Though I gave a 3, I would in fact be happy to raise my score to 8, had the authors addressed the specific points of my review, pointed out my specific errors, or substantially improved their paper based on the comments from all reviewers. One thing I like about ICLR is that it allows revising paper during rebuttal. However, the authors did none of these, and just called my review “lacks specific constructive feedback” and “not aligned with ICLR guidelines”.
> >
> > Overall, this response does not address my concerns, so I will stick to my original judgment. If this response was written by a junior student, then I would suggest they consult their advisors or senior collaborators about how to do research, how to write a paper, and how to interact with peer reviewers.

---

> ### Author Response · Authors · 2023-11-17
> **This response is inappropriately aggressive, and arguably, constitutes bullying**
>
> Dear reviewer ArGT,
>
> This is the supervisor of the student who wrote this paper replying to you now.
>
> First, I want to clarify that I had approved the student's response, and I consider it to be appropriate. I say this as someone who has served as an AC for several ML conference for a few years now. Nothing in the initial response was insulting to you, nor did it say anything unfair. You perhaps disagreed with it, which is fine, and you can and should state so and provide a counter-rebuttal. But claiming that the rebuttal was so bad that the student should, and I quote, "...consult their advisors or senior collaborators about how to do research, how to write a paper, and how to interact with peer reviewers" is completely unfair and inappropriately aggressive. Notably, it was pointed out to you that you were criticising this paper for doing some of the same work as two contemporaneous papers (you stated it was a "known result"). And, as the initial rebuttal noted, that is a violation of ICLR guidelines on your part. You should simply have apologised for violating the guidelines or just clarified that you shared these papers simply for information's sake, not because you are claiming that this paper is not novel and/or should have cited them. That would be fine. Instead, you attacked the graduate student, their work, and their professionalism. This is shameful to see, and quite frankly, smacks of bullying.
>
> Second, it is 100% fair to say that your critiques in point 2 were not about the theory, per se, but about the writing (indeed, you titled the point "On the mathematical writing of this work"). Perhaps "typo" is not quite the right word choice... But, the point of the rebuttal holds: your critiques are critiques on the writing style, not on the theory itself, and arguably, critiques of that nature fall into the category of minor points, as stated in the rebuttal.
>
> Third, I want to end by noting the irony that unlike the initial rebuttal to your review, your reaction was deeply unprofessional. Speaking as an AC, and as a supervisor, I would encourage you to tone down your hostility and act with a greater deal of decorum in future interactions on conference reviews. Please don't needlessly bully students on OpenReview. Behaviour like this degrades our community and weakens our collective ability to effectively train the next generation of scientists.
>
> Sincerely,
>
> Senior author for paper 6318

---

> > ### Comment · Reviewer_ArGT · 2023-11-17
> > **Response**
> >
> > Dear authors or paper 6318,
> >
> > I regret to see that the authors including the senior author, instead of specifically addressing any scientific point in my review, chose to write responses that are so unnecessarily aggressive and so bizarrely insulting. I don't believe that my review or my earlier response is "bullying" in any sense. Like I said, my one and only role is to help the authors improve their work by offering my most honest opinion. Rather, it is really the authors who wrote a quite insulting response to a volunteer reviewer who just wanted to help.
> >
> > I believe that this correspondence has become meaningless because the authors have chosen not to respond to any of my scientific questions, so I will stop here and I will express my most honest opinion to the AC. Thankfully, ICLR is on openreview where everyone can see this correspondence, and I believe that people will have their own fair judgment.
> >
> > Best regards,
> > Reviewer ArGT

---

> ### Author Response · Authors · 2023-11-17
> **Agreed, that this is not productive**
>
> I agree that this is not productive, and there is no need for further back and forth. We have your comments, which were helpful in some places, so thank you for those components. And as noted, everything is on the record for all to see, which I am also happy about. I stand by the claim that the student's initial rebuttal was perfectly professional, and did not insult you, but merely pointed out the fact that you had violated ICLR guidelines (which you did).
>
> I also maintain that the student did not deserve to be spoken down to in the way you did. I think it is a real shame that you are unwilling to recognise and/or reflect on the unnecessary hostility in your reply.
>
> We will also be commenting to the AC, of course.
>
> Sincerely,
>
> Senior author for paper 6318

---

### Official Review · Reviewer_Ajht · 2023-10-28

**Soundness:** 2 fair
**Presentation:** 3 good
**Contribution:** 3 good
**Rating:** 3
**Confidence:** 3

**Summary:**

This paper introduces several enhancements to non-contrastive Self-Supervised Learning (SSL) methods, specifically BarlowTwins and VICReg. The primary claims made are that these methods do not require a high projection dimension, and the utilization of multiple augmentations can enhance performance. The authors provide empirical evidence demonstrating the effectiveness of these improvements on smaller datasets, such as CIFAR10 and STL10.

**Strengths:**

1. The assertion that BarlowTwins and VICReg do not necessitate large projection dimensions constitutes a significant improvement. This assertion is further supported by eigenvalue analysis conducted in the study.

2. The paper conducts a valuable comparison of the utility of multiview techniques in the context of BarlowTwins and VICReg. Experiment results demonstrates the effectiveness of using a multiview approach.

3. The authors offer practical recommendations on how to apply these Self-Supervised Learning (SSL) methods, which enhances the paper's utility for potential users.

**Weaknesses:**

1. The experimental results presented in the paper are less than convincing and appear to involve an unfair comparison. Figure 4, in particular, showcases curves that have not converged. A fair comparison should ensure that all models are optimized and have reached convergence.

2. The utilization of multiview is not a novel concept, and it appears to be an inferior approach when compared to the multi-crop technique employed in SwAV. SwAV has thoroughly investigated this and found that using full views can be less effective due to increased memory overhead. Surprisingly, this paper does not address memory usage and effective computation, which raises questions about the validity of its claims. If each iteration consumes more computational resources and requires additional memory, it is expected to converge faster, making it a critical factor to consider.

3. There seems to be a lack of a clear connection between the discussion of the graphs and the paper's main idea, which can make the paper's arguments less coherent.

**Questions:**

None

---

> ### Author Response · Authors · 2023-11-16
> **Official response to Reviewer Ajht [1/2]**
>
> We appreciate the time and effort the reviewer has dedicated to evaluating our paper. We are glad that the reviewer found our presentation and contribution good (with a score of 3/4), and acknowledge that our work presents “several enhancements to non-contrastive Self-Supervised Learning (SSL) method” and “offer practical recommendations […] which enhances the paper's utility for potential users”. Furthermore, the reviewer agrees that our core insights, specifically that not needing large projection dimensions for non-contrastive SSL, “constitutes a significant improvement” and that the “paper conducts a valuable comparison of the utility of multiview technique.”
>
> Below, we would like to address the concerns raised by the reviewer.
>
> 1. ### On Experimental Results and Convergence:
>
> We thank the reviewer for pointing out this apparent lack of clarity in the result figures. We would like to reiterate that our main claim is that using more augmentations will lead to faster convergence, i.e., learning better features earlier in pretraining. Our current results demonstrate that using multiple augmentations leads to better linear readout performance earlier in training, both in terms of full passes over the dataset (epochs) and wall clock time (see Fig 4). We incorporated the feedback and **ran new experiments where we pretrain each model to 100 epochs** (see updated Fig. 4 & 5) to align with standard practice in SSL pretraining of ResNet50 networks. However, the reviewer’s concerns are valid in settings with sufficient compute budget to pretrain models till convergence and select the best-performing checkpoint. To address this concern, we **have run additional experiments wherein we pretrain ResNet50 models with BarlowTwins on CIFAR-10 up to 400 epochs** and have added these new results (see Appendix A). In line with our claims, using multiple augmentations leads to similar (or even slightly better) performance much earlier in training.
>
> | **Algorithm**                          | **Best Accuracy (%)** | **Epoch of Best Accuracy** |
> |:--------------------------------------:|:---------------------:|:-----------------------------:|
> | Barlow-Twins (2-augs) w/ pdim=256      $\hspace{1cm}$ | 92.04 ± 0.16      $\hspace{1.5cm}$    | 400          $\hspace{1cm}$                 |
> | Barlow-Twins (4-augs) w/ pdim=256      $\hspace{1cm}$ | 92.39 ± 0.17      $\hspace{1.5cm}$    | 340           $\hspace{1cm}$                |
> | Barlow-Twins (8-augs) w/ pdim=256      $\hspace{1cm}$ | 92.64 ± 0.10       $\hspace{1.5cm}$   | 140            $\hspace{1cm}$               |
>
>
> 2. ### Innovation and Utility of Multiview Technique:
>
> We agree with the reviewer that using multiple views is a well-established heuristic in contrastive SSL frameworks like SwAV. However, our core contribution is two fold:
> - We provide theoretical principles that explain why a multiview approach is useful in practice while extending this approach to non-contrastive learning setups like BarlowTwins and VICReg.
> - Furthermore, our results demonstrate that the multiview approach not only yields good performance but is also useful for learning good features from fewer unlabeled samples. As highlighted by Reviewer QPCx our insights present “a strategy that reduces the data needed to learn good representations may benefit the community.”
>
> We would also like to note that SwAV presents a useful empirical approach to use multiple augmentations and would be a special case of our theoretical framework, wherein using crops of smaller size leads to a particular choice of mapping $M$ (and correspondingly operator $T_M$). Nevertheless, we have also **added some new experiments** (see Appendix A) where we adopt a SwAV-style approach, i.e., 2 augmentations of full resolution and additional augmentations of smaller resolution, and demonstrate minor performance gains (at lesser compute cost than full-view augmentations) over the vanilla 2-view approach. While the decreased performance gain (as opposed to using full views for all augmentations) could stem from an implicit bias over the scale of details in learned features from the particular choice of $M$, these results still align with our theoretical conclusions regarding the utility of multiple augmentations. Taken together, we would like to disagree with the reviewer’s claim about the inferiority of our method compared to SwAV and instead present SwAV as a special (compute-efficient) case of our framework.

---

> ### Author Response · Authors · 2023-11-16
> **Official response to Reviewer Ajht [2/2]**
>
> 3. ### Memory Usage and Computation Efficiency:
>
> The reviewer's point on memory usage and computation efficiency is well-taken. Our omission of an in-depth discussion on this aspect was due to space constraints and our primary focus on the theoretical aspects of SSL. However, we agree that this is an important factor and will add a section discussing computational resources and memory usage in the context of our proposed methods. Would such an addition address the reviewer's concerns in this area?
>
> 4. ### Connection Between Graphs and Main Idea:
>
> We thank the reviewer for pointing out this apparent lack of clarity. The main idea of our paper is to make SSL more compute and sample efficient, i.e., to learn good features with a lower compute footprint from fewer unique unlabelled samples. Our results demonstrate that contrary to dominant heuristics for NC-SSL pretraining,  it is possible to **learn good features with smaller projector dimensionality** (Fig. 3), thereby reducing the compute footprint of our models. Furthermore, our theoretical results suggest the **potential of using multiple augmentations to learn good features with fewer passes over the dataset**, i.e., training epochs (Fig. 4) as well as **with fewer unique unlabeled samples** (Fig. 5). Therefore, we believe these results are strongly aligned with the main idea in the paper but are happy to add further clarification in relevant sections. We would greatly appreciate specific feedback from the reviewer on this point.
>
>
> In summary, we would like to thank the reviewer for supporting our paper with high scores on contribution and presentation (3/4) and hope that the above clarifications will improve the reviewer’s confidence in the soundness of our work. We are happy to clarify any other concerns that would potentially improve the reviewer’s score of our paper.

---

> > ### Author Response · Authors · 2023-11-22
> > **Gentle reminder**
> >
> > Dear Reviewer Ajht,
> >
> > We are following up on the rebuttal we submitted in response to your comments on our paper. We are grateful for the time and effort you invested in reviewing our work and your feedback has allowed us to enhance the clarity and strengths of our paper.
> >
> > To reiterate the changes we made in response to your comments, we have conducted additional experiments on pretraining networks for longer epochs as well as using a SwAV-like approach to efficiently leverage multiple augmentations, thereby strengthening our empirical results and further solidifying the paper's contributions.
> >
> > Following our revisions, Reviewer QPCx (who raised similar issues as you) has increased their score to recommend accepting our paper.  As the discussion period deadline is fast approaching, we would like respectfully encourage the reviewer to also update their score if they feel it appropriate, or to clarify any remaining details or additional concerns that prevent the paper being recommended for acceptance.
> >
> > Again, thank you for your valuable input and the time dedicated to reviewing our paper.
> >
> > Best regards,
> >
> > The Authors

---

### Official Review · Reviewer_tALx · 2023-11-09

**Soundness:** 4 excellent
**Presentation:** 4 excellent
**Contribution:** 3 good
**Rating:** 8
**Confidence:** 3

**Summary:**

This paper investiages non-contrastive SSL techniques like BarlowTwins and VICReg from a more foundational perspective. The main theoretical result is that the loss formulation of non-contrastive SSL techniques leads to learning the eigenfunctions of the data covariance kernel that results from the data augmentations used to train the non-contrastive SSL setup. This leads to two concrete practical takeaways: stronger orthogonality constraints allow using smaller projection heads, and using more augmentations of each sample can improve the training as the data-augmentation kernel is better approximated.

**Strengths:**

The paper is clearly written and follows a good structure. The theory is compelling, providing a deeper understanding of something that had previously been made to work by 'engineering tricks'. That the authors are able to provide concrete improvements to two well-known non-contrastive SSL techniques with these insights further rigidifies the value of the theory.

The paper nicely introduces the important terms and relevant works & basic theoretical components, before providing the main result and the corollaries that lead to practical improvements. I only was able to check the main argumentation of the proof in the appendix, which seemed reasonable, and could not into every detail of every step. The existing experimental results seem convincing.

**Weaknesses:**

In page 7, the authors wrote that they train the setup for 2, 4, and 8 augmentations per sample, but Figure 4 only shows results for 2 and 4.

To me it seems a bit mysterious that the main argument seems to be sample-efficiency. I would expect that if training with more augmentations leads to better training, then training with more augmentations (keeping the dataset size fixed) should lead to better final downstream performance. To me it seems like an obvious set of experiments to run, so unless proven otherwise, its absence suggests that the the changes to the nc-SSL objectives do not improve final downstream performance. It would be good to understand this more.

**Questions:**

What happens if you repeat the experiments from Figure 5 with {4, 8} (and even more?) data augmentations but with 100% of the training data?

**Details Of Ethics Concerns:**

-

---

> ### Author Response · Authors · 2023-11-16
> **Official response to Reviewer tALx**
>
> We thank the reviewer for their thorough review and insightful comments. We are glad the reviewer found that our paper “follows a good structure,” presenting compelling theoretical insights and “providing a deeper understanding of something that had previously been made to work by 'engineering tricks'.” We also appreciate your recognition of the paper's core contributions, where we “provide concrete improvements to two well-known non-contrastive SSL techniques.”
>
>
> 1. ### On training with full dataset with 4/8 augmentations :
> We thank the reviewer for this valuable suggestion. Accordingly, we add new experiments with 8 augmentations with Barlow-Twins on the full CIFAR-10 dataset (Appendix A).
>
> | Algorithm                            | #augs=2           | #augs=4           | #augs=8           |
> | :------------------------------------: | :-----------------: | :-----------------: | :-----------------: |
> | Barlow-Twins (CIFAR-10) w/ pdim=256       $\hspace{1cm}$ |    86.43 $\pm$ 0.72   $\hspace{1cm}$ | 91.73 $\pm$ 0.16  $\hspace{1cm}$  | 92.71 $\pm$ 0.19  $\hspace{1cm}$  |
> | Barlow-Twins (CIFAR-10) w/ pdim=8192     $\hspace{1cm}$ |    85.44 $\pm$ 0.54    $\hspace{1cm}$ | 91.40 $\pm$ 0.32  $\hspace{1cm}$  | 92.40 $\pm$ 0.13 $\hspace{1cm}$  |
>
> From the above table, it’s clear that increasing the number of augmentations improves downstream performance in line with our claim. However, using more augmentations also increases the per-epoch time, thereby increasing the overall experiment time. In other words, a tradeoff exists between the performance and the compute time. This tradeoff is further demonstrated in Fig. 6 of our manuscript, where we present a Pareto frontier of performance vs compute time by varying the number of augmentations and the fraction of the dataset. Therefore, given our computing budget, we have restricted our experiments to 8 augmentations.
>
> In summary, we appreciate the constructive feedback and agree that the aspects highlighted for improvement are crucial for a more comprehensive understanding of non-contrastive SSL techniques. We are committed to exploring these areas and believe they will significantly enhance the depth and applicability of our research.

---

### Meta-Review · Area_Chair_zeuF · 2023-12-06

**Metareview:**

This paper studies a class of non-contrastive self-supervised learning methods. Many of these methods are (at least originally) not introduced based on any theoretical formulation, but rather empirically. As a result, they have certain heuristics or rules-of-thumb that are used by practitioners. The authors introduce a theoretical formulation for methods like BarlowTwins and Variance-Invariance-Covariance Regularization. They mine this theory for insights that can produce useful practical advancements.

I view this paper as borderline. There were disagreements between the reviewers, as well as between some of the reviewers and the authors. On the strengths side, the overall thrust of the paper, the resulting insights, and the empirical results were interesting and useful. On the weaknesses side, certain aspects of the theory were either confusing (or incomplete) to me.

For example, Theorem 3.1 is one of the main results of this work, and used to extract the practical insights. The result is a bit confusing as stated, but more critically, it does not appear to have a proof (i.e., there is a sketch in the appendix but without any level of detail).

More generally, the theory needs work to be cleaned up, be made rigorous, etc. I don't doubt that this can be done, since I don't necessarily see fatal flaws. However, as it stands, the paper is not quite there.

I will note that one reviewer describe some (potentially concurrent) work. Because this work may fit within the ICLR timeline for being concurrent, so that the authors did not need to consider it, I did not use it to make the decision. In this metareview more broadly, I avoid using any comparisons to past or concurrent work.

**Justification For Why Not Higher Score:**

The paper needs a number of fixes to establish some of its basic claims, which may not be totally doable in time.

**Justification For Why Not Lower Score:**

N/A

---

### Decision · Program_Chairs · 2024-01-16

Reject